computational biology

STAT1, cancer, mathematical model, apoptosis, optimal control

**Author for correspondence:**
Yangjin Kim
e-mail: ahyouhappy@gmail.com

# Mathematical model of STAT signalling pathways in cancer development and optimal control approaches

Jonggul Lee[1], Donggu Lee[2] and Yangjin Kim[2,3,4]

[1]Pierre Louis Institute of Epidemiology and Public Health, Paris 75012, France
[2]Department of Mathematics, Konkuk University, Seoul 05029, Republic of Korea
[3]Mathematical Biosciences Institute, Columbus, OH 43210, USA
[4]Department of Neurosurgery, Harvard Medical School & Brigham and Women's Hospital, Boston MA 02115, USA

YK, 0000-0002-8905-8481

In various diseases, the STAT family display various cellular controls over various challenges faced by the immune system and cell death programs. In this study, we investigate how an intracellular signalling network (STAT1, STAT3, Bcl-2 and BAX) regulates important cellular states, either anti-apoptosis or apoptosis of cancer cells. We adapt a mathematical framework to illustrate how the signalling network can generate a bi-stability condition so that it will induce either apoptosis or anti-apoptosis status of tumour cells. Then, we use this model to develop several anti-tumour strategies including IFN-β infusion. The roles of JAK-STATs signalling in regulation of the cell death program in cancer cells and tumour growth are poorly understood. The mathematical model unveils the structure and functions of the intracellular signalling and cellular outcomes of the anti-tumour drugs in the presence of IFN-β and JAK stimuli. We identify the best injection order of IFN-β and DDP among many possible combinations, which may suggest better infusion strategies of multiple anti-cancer agents at clinics. We finally use an optimal control theory in order to maximize anti-tumour efficacy and minimize administrative costs. In particular, we minimize tumour volume and maximize the apoptotic potential by minimizing the Bcl-2 concentration and maximizing the BAX level while minimizing total injection amount of both IFN-β and JAK2 inhibitors (DDP).

# 1. Introduction

Cancer, including lung cancer, is the most fatal killer in the world [1,2]. Comprehensive understanding of signalling networks of

**Figure 1.** A schematic diagram for a proposed network of apoptosis signalling in the presence of IFN-β/JAK2. (*a*) Low IFN-β and high JAK2 levels increase STAT3 and Bcl-2 and suppress STAT1 and BAX, maintaining anti-apoptosis status. (*b*) High IFN-β and decreased JAK2 initiate phenotypical transition from anti-apoptosis to apoptosis of cancer through reversed regulation of each module.

oncogene and tumour suppressors [3–6] in cancer cells can play a significant role in developing anti-cancer drugs [7–9]. Various types of transcription factors function in a coordinated fashion to regulate cell growth, cell division, cell death and cell migration [10–12]. In this work, we focus on the STAT family which was shown to suppress or promote tumour growth [13,14]. There are various subtypes of STAT family including STAT1, -2,-3,-4,-5 (STAT5A and STAT5B) and -6 [15]. Lack of STAT1 indicates a selective signalling defect in response to interferons (IFN). While STAT1 (tumour suppressor) suppresses the aggressive invasion and cellular growth of tumour cells [15–18], STAT3 (oncogene) regulates multiple biological functions such as suppression of apoptosis, cell growth and invasion [15,16,19]. Relative balance between STAT1 and STAT3 levels in cancer cells determines two different dichotomous states: (i) an apoptosis progression and (ii) an anti-apoptosis state (inactivation of cell death program). See figure 1. Various cytokines and growth factor receptors may initiate the JAK/STAT network to target genes. Bcl-2 is a well-recognized gate-keeper, preventing the cellular death of cancer cells by inhibiting BAX [20]. BAX, widely known as a pro-apoptotic factor, represents an opposing function at the last signal step of the programmed cell death mechanism, i.e. apoptosis [21,22]. The apoptotic process is mediated by the suppression of Bcl-2 as well as the activation of BAX [21,23]. Reduced DNA binding ability of STAT3 causes dynamic changes in the expression of anti-apoptotic Bcl-2 (decreased Bcl-2) and pro-apoptotic BAX proteins (increased BAX), leading to induction of apoptosis [24]. Type I interferons and receptors mediate the downstream signals via TYK2, JAK1 and JAK2, and through the phosphorylated STATs. Inhibitors of interferon signals, such as suppressors of the cytokine signal protein family, can control the activity of STAT1 and STAT3 primarily by engaging in the adjustment of the negative feedback of JAK2-mediated signal network [25]. Despite previous studies of apoptotic signalling, fundamental mechanism of the JAK-IFN-β-mediated apoptosis processes is poorly understood. However, translational studies with experimental data [26] support the considerable benefits of apoptosis-based therapy [27]. Qualitative analysis may contribute to fundamental understanding of this complex system. In particular, a new approach may identify the key functions and regulation of both JAK and IFN-β-mediated STATs in the apoptosis pathways within cancer cells.

Mathematical modelling is a useful tool in revealing the fundamental mechanism in various cancers [5,28–31], interactions with other cells [9,32–34] including immune cells [7,35,36], cellular invasion [37], chemotherapy of cancer [35,38,39], apoptosis mechanism [6,35,40,41], and specific signalling pathways [8] such as JAK-STAT [42,43], MYC-p53 [4] and microRNAs [3]. For example, mathematical models of Bcl-2 signalling networks illustrated the importance of molecular play including intrinsic Bcl-2 apoptosis pathway [44–48], bistability in apoptosis [49], interaction between p53 and Bcl-2 [50], VEGF-

Bcl-2 in angiogenesis [51,52] and MOMP regulation in pattern recognition [53]. See reviews in [54–57] for systems-based approaches of Bcl-2 and cell-death program. In particular, optimal control approaches are used to identify the optimal schedule of anti-cancer drugs targeting stromal/immune cells and various signalling pathways [58–61]. The fundamental mechanism of the JAK-STAT-mediated cancer cell killing is still poorly understood. To our knowledge, no mathematical study has investigated the underlying mechanisms of JAK-STAT mediation of apoptosis in cancer cells. We have developed a mathematical model of JAK-STAT-mediated apoptosis pathways in regulation of tumour growth and cancer cell killing. We investigate the optimal dose schedule of anti-cancer drugs by an optimal control theory.

Figure 1 shows a schematic diagram for a proposed network of apoptosis signalling in the presence of IFN-β/JAK2. The network consists of a system of ordinary differential equations (ODEs) involving eight variables: concentrations of STAT1, STAT3, Bcl-2, BAX, IFN-β, JAK2 and DDP and tumour volume. In this work, we investigate (i) unexplored structure of the STAT-JAK2-Bcl-2-BAX signalling pathways, (ii) how changes in IFN-β, STATs and JAK2 affect cancer progression, (iii) development of optimized treatment scheme in a polymedicine approach (IFN-β + JAK2 inhibitor). We found that JAK2 and mutual antagonism between STAT1 and STAT3 play a major role in regulation of the apoptosis and anti-apoptotic status in cancer cells, thus tumour growth dynamics, and obtained the optimal injection strategies of both JAK2 inhibitors and IFN-β by minimizing costs and maximizing anti-tumour efficacy through an optimal control theory.

## 2. Methodology

### 2.1. Mathematical model

*Intracellular module (STAT1, STAT3, Bcl-2 and BAX)*: to consider the key pathways of apoptotic cell death in a mathematical approach, we simplified the complex network to a key network shown in figure 2*a*. Conventionally, the kinetic notations of hammerheads and solid arrows in a signalling network represent inhibition and induction, respectively. Let the variables $\bar{S}_1$, $\bar{S}_3$, $\bar{B}$ and $\bar{X}$ be concentrations of STAT1, STAT3, Bcl-2 and BAX at time $\bar{t}$, respectively.

The scheme includes autocatalytic activities, nonlinear activation or inhibition, mutual inhibition between STAT1 and STAT3, and clearance/decay. In this work, we ignore any spatial effects on dynamics of a given system. In general, the mass balance of given intracellular variable $y_i = y_i(t)$, $(i = 1, \ldots, N)$ is used to derive the governing equation

$$\frac{dy_i}{dt} = f_i(\mathbf{y}) + g_i(\mathbf{y}) - h_i(\mathbf{y}), \tag{2.1}$$

where $\mathbf{y} = (y_1, y_2, \ldots, y_N)$, the function $f_i(y)$ represents the source, $g_i(y)$ represents inhibition, and $h_i(y)$ represents outflux due to natural decay, i.e. $h_i(y) = \mu_i y_i$, where $\mu_i$ is the decay rate. The source function $f_i(y)$ can be described below based on biological observations. A fractional form for the inhibition term in equation (2.1) was chosen as the qualitative representation of negative feedbacks in this work. Specifically, we use the form

$$g_i(\mathbf{y}) = \frac{\zeta_1 \zeta_2^n}{\zeta_2^n + \alpha_i F(y_j)} \tag{2.2}$$

for autocatalytic activity with the inhibition process of the intracellular variable $y_i$ by another intracellular variable $y_j$ $(i \neq j)$, where $\zeta_1$, $\zeta_2$ are constants, the parameter $\alpha_i$ represents the inhibition strength along with the amount of the variable $y_j$ via a function $F(y_j)$ $(\mu_i, \zeta_1, \zeta_2, \alpha_i \in \mathbb{R}^+, n \in \mathbb{Z}^+)$. In the absence of source, this inhibition term with the decay term, $-\mu_i y_i$ provides the baseline concentration $y_i^* \approx \zeta_1/\mu_i$ of the given molecule $y_i$ at a steady state when the inhibition strength $F(y_j)$ is small or zero. (When $f_i(\mathbf{y}) \neq 0$, the baseline becomes $y_i^* \approx (f_i + \zeta_1)/\mu_i$.) The relative balance between the source term and inhibition strength from $y_j$ essentially determines the concentration of the molecule $y_i$. Thus, by comparing the simulated $y_i$ level with experimental data in the presence and absence of the inhibitory molecule $y_j$ in the system, one can build a mathematical model in equation (2.1) with the consistent, up- or downregulated $y_i$. Several studies [3,5,35,62] have shown that this fractional form for the negative feedbacks may reproduce the analytic structure of genetic networks (positive and negative feedbacks) and qualitative dynamics such as bi-stability with experimental validation. Other forms of negative feedbacks (e.g. one based on chemical reactions) have been used

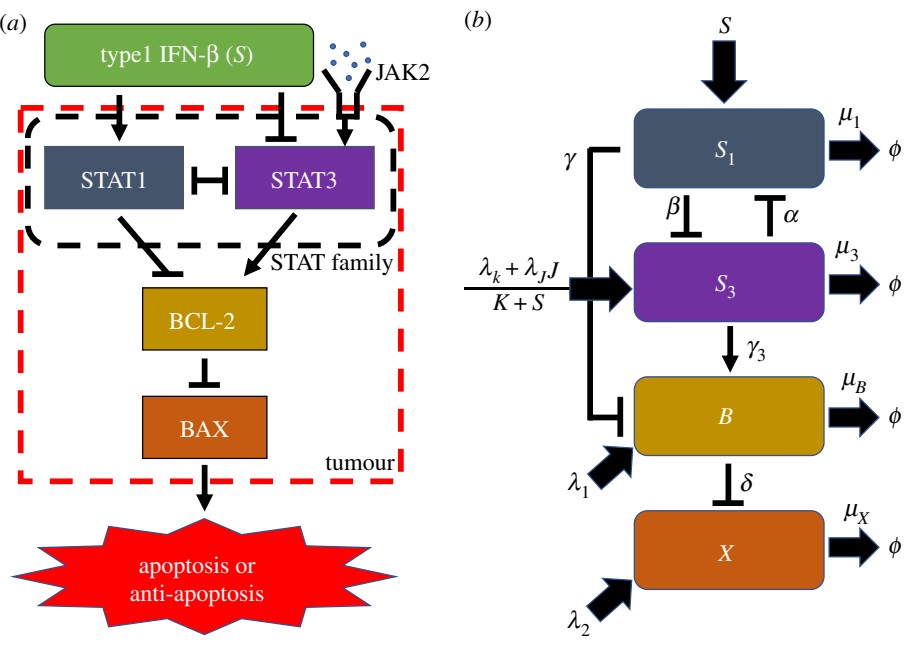

**Figure 2.** A schematic diagram of the apoptosis signalling network in figure 1. (*a*) Key signalling network of apoptosis involving STAT1, STAT3, Bcl-2 and BAX in response to IFN-β and JAK2. (*b*) The corresponding mathematical model: levels of STAT1 and STAT3, and activity of their target Bcl2, and BAX were represented by '$S_1$', '$S_3$', '$B$' and '$X$', respectively.

in the literature [32,63]. Then, the mass balance of the concentrations of STAT1 ($\bar{S}_1$), STAT3 ($\bar{S}_3$), Bcl-2 ($\bar{B}$) and BAX ($\bar{X}$) gives us

$$\frac{d\bar{S}_1}{d\bar{t}} = \underbrace{f_1(s)}_{\text{source}} + \underbrace{\frac{a_1 a_2^2}{a_2^2 + a_3 F_1(\bar{S}_3)}}_{\bar{S}_3 \dashv \bar{S}_1} - \underbrace{\mu_{S_1} \bar{S}_1}_{\text{decay}}, \tag{2.3}$$

$$\frac{d\bar{S}_3}{d\bar{t}} = \underbrace{f_2(s,j)}_{\text{source}} + \underbrace{\frac{a_4 a_5^2}{a_5^2 + a_6 F_2(\bar{S}_1)}}_{\bar{S}_1 \dashv \bar{S}_3} - \underbrace{\mu_{S_3} \bar{S}_3}_{\text{decay}}, \tag{2.4}$$

$$\frac{d\bar{B}}{d\bar{t}} = \underbrace{f_3}_{\text{source}} + \underbrace{\frac{a_7 a_8^2}{a_8^2 + a_9 F_3(\bar{S}_1)}}_{\bar{S}_1 \dashv \bar{B}} + \underbrace{\lambda_{STAT3} \bar{S}_3}_{\bar{S}_3 \to \bar{B}} - \underbrace{\mu_{Bcl2} \bar{B}}_{\text{decay}} \tag{2.5}$$

and

$$\frac{d\bar{X}}{d\bar{t}} = \underbrace{f_4}_{\text{source}} + \underbrace{\frac{a_{10} a_{11}^2}{a_{11}^2 + a_{12} F_4(\bar{B})}}_{\bar{B} \dashv \bar{X}} - \underbrace{\mu_{BAX} \bar{X}}_{decay}, \tag{2.6}$$

where $s$, $j$ are concentrations of IFN-β and JAK2, respectively. The rate of changes in STAT1 involves the signal source from IFN-β via a function $f_1(s)$, autocatalytic activity with inhibition from STAT3 ($\bar{S}_3 \dashv \bar{S}_1$) and natural decay at a rate $\mu_{S_1}$. In particular, the general form in equation (2.2) is used for the autocatalytic activity/inhibition in the second term on the right-hand side (RHS) of equation (2.3) with the autocatalytic activity parameter $a_1$, the Hill-type inhibition saturation constants $a_2$, and inhibition strength $a_3$. Consistent forms and parameter notations were used for autocatalytic activity with inhibition of STAT3 ($a_4$, $a_5$, $a_6$), Bcl-2 ($a_7$, $a_8$, $a_9$) and BAX ($a_{10}$, $a_{11}$, $a_{12}$) in the second terms in equations (2.4)–(2.6). In a similar fashion, STAT3 in equation (2.4) undergoes the signalling from both JAK and IFN-β via a function $f_2(s, j)$, autocatalytic activity with inhibition from STAT1 ($\bar{S}_1 \dashv \bar{S}_3$) and natural decay at a rate $\mu_{S_3}$. On the other hand, Bcl-2 in equation (2.5) is regulated by the signal source at a fixed rate $f_3$, autocatalytic activity with inhibition from STAT1 ($\bar{S}_1 \dashv \bar{B}$), upregulation from STAT3 ($\bar{S}_3 \to \bar{B}$) at a rate $\lambda_{STAT3}$, and natural decay at a rate $\mu_{Bcl2}$. Finally, BAX in equation (2.6) is regulated by the signal source at a fixed rate $f_4$, autocatalytic activity with inhibition from STAT1 ($\bar{B} \dashv \bar{X}$), and natural decay at a rate $\mu_{BAX}$. We set $\mu_{S_1} = \mu_{S_3}$ due to the same half-life of STAT1 and STAT3 (electronic supplementary material, File).

In equation (2.3), the high concentration of IFN-β ($s$) upregulates the STAT1 level through the positive function $f_1(s)$, while the high concentration of STAT3 inhibits the STAT1 level through the positive

function $F_1(\bar{S}_3)$. In other words, we have mathematical conditions: $\partial f_1/\partial s > 0$, $\forall s \geq 0$ and $\partial F_1/\partial \bar{S}_3 > 0$, $\forall \bar{S}_3 \geq 0$. In a similar fashion, the function $f_2(s, j)$ in equation (2.4) indicates the upregulation of the STAT3 level through JAK as well as suppression of the STAT3 level by the IFN-β. STAT1-mediated suppression of STAT3 is expressed by the function $F_2(\bar{S}_1)$. On the contrary, STAT3 activity is partially induced by the signal $j$. One must also have $\partial f_2/\partial s < 0$, $\partial f_2/\partial j > 0$, $\partial F_2/\partial \bar{S}_1 > 0$, $\partial F_3/\partial \bar{S}_1 > 0$, $\partial F_4/\partial \bar{B} > 0$ for all non-negative $s$, $j$, $\bar{S}_1$, $\bar{B}$. Based on biological assumptions (figure 2a), we assume that

$$f_1(s) = \lambda_{IFN\beta}s, \ \ f_2(s, j) = \frac{K_2 + \lambda_{JAK}j}{K_1 + \lambda_{IFN\beta 2}s},$$

$$F_1(\bar{S}_3) = \bar{S}_3^{\ 2}, \ \ F_2(\bar{S}_1) = \bar{S}_1^{\ 2}, \ \ F_3(\bar{S}_1) = \bar{S}_1^{\ 2}, \ \ F_4(\bar{B}) = \bar{B}^2,$$

(2.7)

where $\lambda_{IFN\beta}$ is the source of STAT1 from IFN-β, $\lambda_{IFN\beta 2}$ is a source from IFN-β, and $\lambda_{JAK}$ is a source from JAK2. We use a non-dimensionalization formula as follows:

$$
\left.
\begin{aligned}
&t = \mu_{S_1}\bar{t}, \ \ S_1 = \frac{\bar{S}_1}{S_1^*}, \ \ S_3 = \frac{\bar{S}_3}{S_3^*}, \ \ B = \frac{\bar{B}}{B^*}, \ \ X = \frac{\bar{X}}{X^*}, \ \ \mu_B = \frac{\mu_{Bcl2}}{\mu_{S_1}}, \ \ \mu_X = \frac{\mu_{BAX}}{\mu_{S_1}}, \\
&k_1 = \frac{a_1}{\mu_{S_1}S_1^*}, \ \ k_2 = a_2, \ \ \alpha = a_3(S_3^*)^2, \ \ k_3 = \frac{a_4}{\mu_{S_1}S_3^*}, \ \ k_4 = a_5, \ \ \beta = a_6(S_1^*)^2, \\
&\lambda_1 = \frac{f_3}{\mu_{S_1}B^*}, \ \ k_5 = \frac{a_7}{\mu_{S_1}B^*}, \ \ k_6 = a_8, \ \ \gamma = a_9(S_1^*)^2, \ \ \lambda_3 = \frac{\lambda_{STAT3}S_3^*}{\mu_{S_1}B^*}, \\
&\lambda_2 = \frac{f_4}{\mu_{S_1}X^*}, \ \ k_7 = \frac{a_{10}}{\mu_{S_1}X^*}, \ \ k_8 = a_{11}, \ \ \delta = a_{12}(X^*)^2, \ \ \lambda_{S_1} = \frac{\lambda_{IFN\beta}S^*}{\mu_{S_1}S_1^*}, \ \ S = \frac{s}{S^*}, \\
&\lambda_k = \frac{K_2}{\mu_{S_1}S_3^*}, \ \ \lambda_J = \frac{\lambda_{JAK}J^*}{\mu_{S_1}S_3^*}, \ \ J = \frac{j}{J^*}, \ \ K = K_1, \ \ \lambda_{S_2} = \lambda_{IFN\beta 2}S^*.
\end{aligned}
\right\}
$$

(2.8)

and

Then, we have dimensionless governing equations

$$\frac{dS_1}{dt} = \underbrace{\lambda_{S_1}S}_{source} + \underbrace{\frac{k_1k_2^2}{k_2^2 + \alpha S_3^2}}_{S_3 \dashv S_1} - \underbrace{S_1}_{decay},$$

(2.9)

$$\frac{dS_3}{dt} = \underbrace{\frac{\lambda_k + \lambda_J J}{K + \lambda_{S_2}S}}_{source} + \underbrace{\frac{k_3k_4^2}{k_4^2 + \beta S_1^2}}_{S_1 \dashv S_3} - \underbrace{S_3}_{decay},$$

(2.10)

$$\frac{dB}{dt} = \underbrace{\lambda_1}_{source} + \underbrace{\frac{k_5k_6^2}{k_6^2 + \gamma S_1^2}}_{S_1 \dashv B} + \underbrace{\lambda_3 S_3}_{S_3 \to B} - \underbrace{\mu_B B}_{decay}$$

(2.11)

and

$$\frac{dX}{dt} = \underbrace{\lambda_2}_{source} + \underbrace{\frac{k_7k_8^2}{k_8^2 + \delta B^2}}_{B \dashv X} - \underbrace{\mu_X X}_{decay}.$$

(2.12)

The mathematical representation of the kinetic network in a dimensionless form is shown in figure 2b. Note that the programmed cell death of cancer cells (i.e. apoptosis) occurs when BAX is upregulated and the anti-apoptotic agent, i.e. the gate keeper Bcl-2, is downregulated. Therefore, in our modelling framework, the *apoptosis* process is turned on when the level of Bcl-2 is smaller than a threshold ($th_B$) and BAX activity is larger than another one ($th_X$); in other words, when the condition $\{(B, X) : B < th_B, \ X > th_X\}$ is satisfied, as suggested in experiments and modelling works [6,49,64–68]. The threshold values were set based on biological observation [49,64–68] and dynamical system of equations (2.9)–(2.12).

*Tumour volume (T(t))*: Bcl-2 and BAX are crucial in the mutual antagonism of cell death programs through IFN-β and JAK2 in a tumour microenvironment (TME). STAT1 is shown to suppress cancer growth [15–18]. We assume that tumour cell killing is regulated by relative balance between Bcl-2 and BAX, and tumour growth is suppressed by STAT1. Various types of mathematical models for tumour growth were suggested: logistic growth [69], Gompertz growth [70] and other nonlinear models [71]. Especially, logistic growth with/without growth factors was observed in the experiments [7,34,72–74]. Mathematical models (either ODEs, PDEs or multi-scale types) were designed for comparison with experiments [5–7,33–36,62,72–79].

We have the following assumptions: (i) growth of tumour cells follows the logistic growth with a carrying capacity $T_0$ and STAT1-mediated saturation and (ii) the tumour cells are killed by apoptosis at a rate $\mu_T$. In particular, inhibition of tumour growth by STAT1 in TME [17] is modelled by a Hill type function, $1 - (k_9 S_1^2/(k_{10}^2 + S_1^2))I_{\{B<th_B,\, X>th_X\}}$, where $k_9$ is inhibition strength by STAT1 ($k_9 \leq 1$), $k_{10}$ is a Hill coefficient, and $I_{\{B<th_B,\, X>th_X\}}$ is an indicator function, giving 1 when the *apoptosis* condition ($B < th_B$, $X > th_X$) is satisfied with threshold values of Bcl-2 ($th_B$) and BAX ($th_B$), and 0 otherwise. Thus, the governing equation for the tumour volume ($T$) is

$$\frac{dT}{dt} = \underbrace{r\left(1 - \frac{k_9 S_1^2}{k_{10}^2 + S_1^2}I_{\{B<th_B,\, X>th_X\}}\right)T\left(1 - \frac{T}{T_0}\right)}_{\text{growth}} - \underbrace{\mu_T T I_{\{B<th_B,\, X>th_X\}}}_{\text{apoptosis}}. \tag{2.13}$$

Here, the first term on RHS of equation (2.13) represents the STAT1-controlled growth of tumour cells. We note that the inhibition part in the middle of the first term $(1 - (k_9 S_1^2/(k_{10}^2 + S_1^2))I_{\{B<th_B,\, X>th_X\}}) \geq 0$, $\forall S_1$, due to our assumption $k_9 \leq 1$. In particular, we set $k_9 = 1$. On the other hand, the second term represents the programmed cell death of tumour cells when the intracellular signalling induces the apoptosis in response to external stimuli such as IFN-β.

*IFN-β (S(t))*: Experimental studies [80] on a combination (IFN-β + another drug) therapy illustrated the effectiveness of the therapy on inhibiting cancer progression [81–84] as well as promoting the immune reactions [85]. For example, the population and immune activities of T cells were notably enhanced after IFN-β injection [86]. IFN-β-based drugs, such as Avonex, Reif and CinnoVex, are well-known anticancer drugs that are administered by intramuscular injection. In our model, type I interferons (IFN − $\alpha$, -β) are injected at a rate $u_S$ for tumour cell killing. Thus, the governing equation for IFN-β ($S$) is

$$\frac{dS}{dt} = \underbrace{u_S(t)}_{\text{injection}} - \underbrace{\mu_S S}_{\text{decay}}. \tag{2.14}$$

The first and second terms on RHS of equation (2.14) represent the injection of IFN-β via a function $u_S(t)$ and decay process at a rate $\mu_S$, respectively. Here, $u_S = u_S(t)$ can be a constant or function of time. In this work, we consider three methods of injection: (i) constant injection, (ii) alternating injection and (iii) optimally controlled injection by optimal control theory.

*JAK2 (J(t)) and DDP (D(t))*: since the JAK family stimulates upregulation of STAT3, which then promotes anti-apoptosis pathways. We introduce the JAK2 inhibitor drug called cisplatin (DDP). DDP is an anti-cancer drug that was developed for effective chemotherapy and widely adapted as a first-choice medicine for cancer [87,88]. The governing equations for JAK2 ($J$) and DDP ($D$) are

$$\frac{dJ}{dt} = \underbrace{J_s}_{\text{source}} - \underbrace{\gamma_D D J}_{\text{degradation}} - \underbrace{\mu_J J}_{\text{decay}}, \tag{2.15}$$

and

$$\frac{dD}{dt} = \underbrace{u_D(t)}_{\text{injection}} - \underbrace{\mu_D D}_{\text{decay}}. \tag{2.16}$$

JAK2 in equation (2.15) undergoes production at a rate $J_s$, the DDP-mediated degradation of JAK by DDP ($\gamma_D$), and decay process at a rate $\mu_J$ in the first, second and third terms on RHS, respectively. The first and second terms on RHS in equation (2.16) represent the time-dependent injection of DDP via a function $u_D(t)$ and decay process at a rate $\mu_D$, respectively.

A dimensional version of equations of tumour volume, IFN-β, JAK2 and DDP corresponding to equations (2.13)–(2.16) was introduced and non-dimensionalization was performed in the electronic supplementary material. Matlab (Mathworks) software was used for computational results of the mathematical model and optimal control problems. See table 1 for parameter values in equations (2.9)–(2.16) in a dimensionless form. Parameter values in the dimensional form are listed in table S1 in the electronic supplementary material. Since our mathematical model contains many known and unknown parameter values, we provided parameter estimation in the electronic supplementary material, which is a necessary step toward fundamental and deep understanding of the dynamical process of the mathematical model. Parameter values are calculated based on empirical data such as half-life or estimated by fitting to experimental observation based on the mathematical structure of our model.

**Table 1.** Parameters of the mathematical model.

| parameter | description | value | refs |
|---|---|---|---|
| intracellular modules | | | |
| $S$ | IFN-β signalling | 0–1.0 | [89], Est |
| $k_1$ | autocatalytic production rate (STAT1 module) | 4.0 | Est |
| $k_2$ | Hill-type coefficient (STAT1 module) | 1.0 | Est |
| $\alpha$ | inhibition strength of STAT1 by STAT3 | 1.5 | Est |
| $\lambda_k$ | signalling source of STAT3 | 1.0 | Est |
| $\lambda_J$ | induction rate of STAT3 by JAK2 | 4.0 | Est |
| $J$ | JAK2 signalling level | 0–1.0 | Est |
| $K$ | inhibition parameter | 5.0 | Est |
| $k_3$ | autocatalytic production rate (STAT3 module) | 4.0 | Est |
| $k_4$ | Hill-type coefficient (STAT3 module) | 1.0 | Est |
| $\beta$ | inhibition strength of STAT3 by STAT1 | 1.0 | Est |
| $\mu_3$ | relative decay rate of STAT3 | 1.0 | [90,91] |
| $\lambda_1$ | signalling source of Bcl-2 | 0.2 | Est |
| $k_5$ | autocatalytic production rate (Bcl-2 module) | 1.0 | Est |
| $k_6$ | Hill-type coefficient (Bcl-2 module) | 1.0 | Est |
| $\gamma$ | inhibition strength of Bcl-2 by STAT1 | 1.0 | Est |
| $\lambda_3$ | signalling from STAT3 | 1.2 | Est |
| $\mu_B$ | relative decay rate of Bcl-2 | 1.2 | [92,93] |
| $\lambda_2$ | signalling source of BAX | 0.2 | Est |
| $k_7$ | autocatalytic production rate (BAX module) | 4.0 | Est |
| $k_8$ | Hill-type coefficient (BAX module) | 1.0 | Est |
| $\delta$ | inhibition strength of BAX by Bcl-2 | 1.0 | Est |
| $\mu_X$ | relative decay rate of BAX | 5.0 | [94,95] |
| threshold | | | |
| $S_1^{th}$ | threshold of STAT1 | 1.8 | Est |
| $S_3^{th}$ | threshold of STAT3 | 1.3 | Est |
| $B^{th}$ | threshold of Bcl-2 | 1.44 | Est |
| $X^{th}$ | threshold of BAX | 0.3 | Est |
| tumour module | | | |
| $r$ | growth rate of tumour cells | 0.12 | [80] |
| $k_9$ | inhibition parameter of STAT1 growth | 1.0 | [80] |
| $k_{10}$ | inhibition parameter of STAT1 growth | 10 | [80] |
| $T_0$ | carrying capacity of a tumour | 100 | [80] |
| $\mu_T$ | killing rate of tumour cells by apoptosis | 0.1 | [80] |
| therapeutics | | | |
| $\mu_S$ | decay rate of IFN-β | 4.8 | [35,96] |
| $J_s$ | source of JAK2 | 1.3 | [97] |
| $\mu_J$ | decay rate of JAK2 | 1.3 | [97] |
| $\gamma_D$ | degradation rate of JAK2 by DDP | 1.0 | Est |
| $\mu_D$ | decay rate of DDP | 10 | [98,99] |

(*Continued.*)

| parameter | description | value | refs |
|---|---|---|---|
| reference value | | | |
| $S_1^*$ | STAT1 concentration | 2.43 µg ml$^{-1}$ | [100] |
| $S_3^*$ | STAT3 concentration | 1.38 µg ml$^{-1}$ | [100] |
| $B^*$ | Bcl-2 concentration | 10 nM | [101] |
| $X^*$ | BAX concentration | 351 µM | [102] |
| $S^*$ | IFN-β concentration | 10 ng ml$^{-1}$ | [89] |
| $J^*$ | JAK2 concentration | 2.8 n M | [103] |
| $D^*$ | DDP concentration | 10 µg ml$^{-1}$ | [104] |
| $T^*$ | tumour volume | 100 mm$^3$ | [80] |

*Est = estimated.

## 2.2. Optimal control strategies

The optimal control theory was used to find an optimal injection profile of drugs that minimizes the tumour volume by controlling the cell-death program, while the amount of the drugs is minimized. Two control variables, $u_S(t)$ in equation (2.14) and $u_D(t)$ in equation (2.16), are the sources of IFN-β and cisplatin (DDP), respectively. An objective is to find the optimal dose and sum of the two drugs over time for the minimal tumour size. Thus, this strategy leads to an objective function as follows [58,105,106]:

$$\mathcal{J}(u_S, u_D) = \min_{u_S, u_D} \int_{t_s}^{t_e} A_1(T(t) - \bar{T})^2 + A_2 B^2 - A_3 X^2$$
$$+ C_1 u_S(t) + C_2 u_D(t) + C_3 u_S(t)^2 + C_4 u_D(t)^2 \, \mathrm{d}t. \quad (2.17)$$

Here, $T(t)$ and $(\bar{T})$ denote the concentration and desired concentration of tumour, respectively. Parameters $A_1$, $A_2$ and $A_3$ are weight constants for the concentration of tumour, Bcl-2 and BAX, respectively. We used quadratic forms to simplify analysis with the convexity properties which are common in control problems in biological models [106]. For the controls in the integrand, we added linear terms to regularize the amount of drug used. In general, linear controls are more meaningful biologically than quadratic forms, but it is more difficult to analyse the system mathematically. Weight for each control is provided by parameters $C_1$, $C_2$, $C_3$, $C_4$. Linear ($u_S(t)$) and quadratic ($u_S(t)^2$) forms in equation (2.17) represent the costs. Note that in the optimal control problem not only is tumour concentration reduced to a certain level but also Bcl-2 (BAX) is minimized (maximized) to induce apoptosis, thus to suppress tumour growth. In most numerical simulations, we set the desired tumour volume to be 50% of the control case that both controls are not used [105,106]. For comparison, we set two control injection schedules: alternating injection and constant injection. The injection rate of IFN-β and DDP in alternating strategy is 5.2344 and 53.4109, respectively. The injection rate of IFN-β and DDP in the constant strategy is 4.1875 and 24.9251, respectively. Note that in all cases, we fixed the total amount of IFN-β and DDP. To obtain the numerical solutions of the control problems, we used the forward–backward sweep method which is based on shooting methods to solve boundary value problems [105].

# 3. Results and discussion

## 3.1. Characterization of apoptosis and anti-apoptosis state

We investigate dynamical properties of given intracellular module, equations (2.9)–(2.12), in the absence of JAK2 (i.e. $J = 0$ in equation (2.10)). Figure 3 shows various cell fates of cancer cells in distinct IFN-β conditions (low (A), transitional (B), high (C)). The equilibrium point (steady state (SS)) is marked as a circle. The steady state of STAT1 in the system (2.9)–(2.12) can be expressed in terms of the

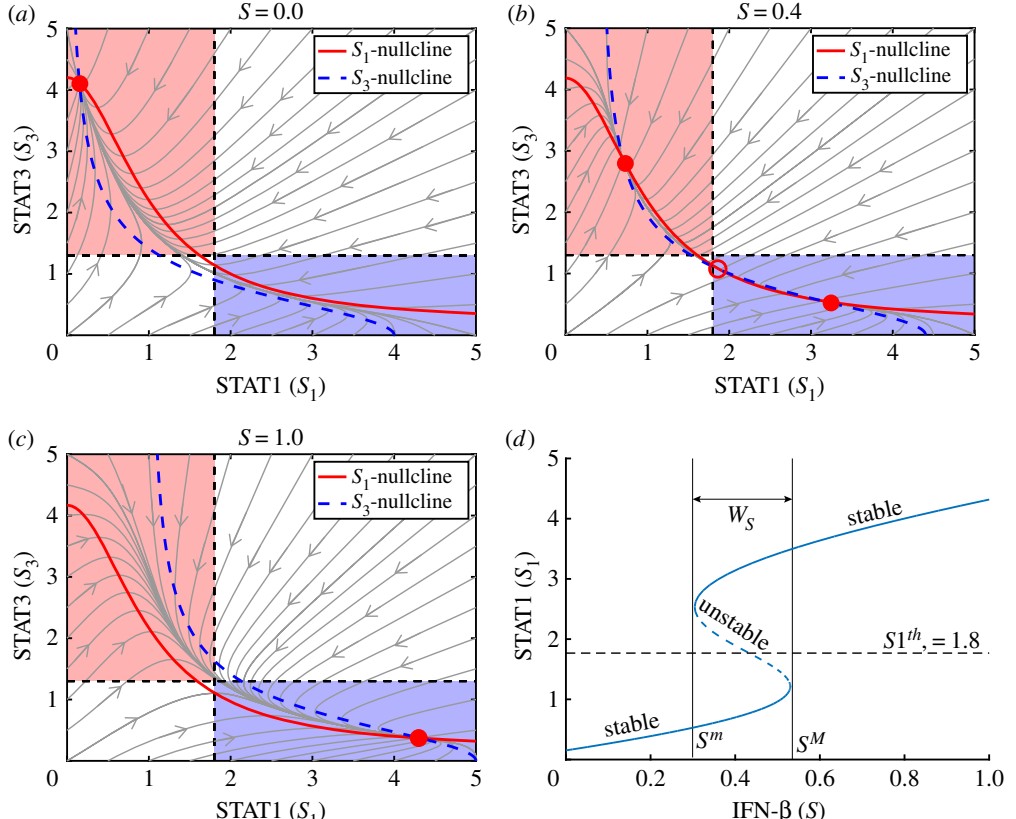

**Figure 3.** Dynamics of intracellular (STAT1-STAT3-Bcl2-BAX) module in the absence of JAK2. (*a–c*) Solution flow of the system (2.9)–(2.12) in the $S_1 - S_3$ set when $S = 0$ (*a*), 0.4 (*b*) and 1.0 (*c*). Filled circle = *stable* equilibrium, empty circle = *unstable* equilibrium. Blue region = upregulation of STAT1 + downregulation of STAT3, pink region = downregulation of STAT1 + upregulation of STAT3. (*d*) Bifurcation curve of STAT1. $Y - axis$ = equilibrium. $W_S = [S^m, S^M]$ = a window of bi–stabilty.

IFN-β level ($S$) as shown in figure 3*d* ($S_1 = S_1(S)$ as a hysteresis). Figure 4*a–c* shows the corresponding bifurcation curves of STAT3 ($S_3 = S_3(S)$), Bcl-2 ($B = B(S)$) and BAX ($X = X(S)$), respectively, implying the *stable* lower/upper SS curves as well as the *unstable* mid-curve in figure 4*d*. By taking the thresholds ($S_1^{th}$, $S_3^{th}$, $B^{th}$, $X^{th}$) of the level of STAT1, STAT3, Bcl-2 and BAX, the *anti-apoptotic* ($\mathbb{P}_t$) and *apoptotic* ($\mathbb{P}_a$) status can be defined as

$$\mathbb{P}_t = \{(B, X) \in \mathbb{R}^2 : B > B^{th}; X < X^{th}\} \tag{3.1}$$

and

$$\mathbb{P}_a = \{(B, X) \in \mathbb{R}^2 : B < B^{th}; X > X^{th}\}. \tag{3.2}$$

Figure 4*d* illustrates the anti-apoptotic (low BAX, high Bcl-2) and apoptotic (high BAX, low Bcl-2) modes in a $B - X$ phase. A small IFN-β amount ($S=0$) causes unique *stable* equilibrium where activities of STAT1 and BAX are suppressed while activities of STAT3 and Bcl-2 are enhanced (figure 3*a*). This leads to the anti-apoptosis status ($\mathbb{P}_t$; figure 4*d*). This $\mathbb{P}_t$ system is maintained to the critical position of the bifurcation branch near $S=0.53$ as $S$ increases. Passing the mentioned critical position, STAT1 activity creeps up to the stable upper arm, leading to the apoptosis phase ($\mathbb{P}_a$; figure 4*d*) where activities of STAT1 and BAX are upregulated while activities of STAT3 and Bcl-2 are downregulated (figure 3*c*).

In the middle interval of IFN-β ($W_S = [S^m, S^M] = [0.3, 0.53]$; bi-stable), the dynamics adapts multiple (3) equilibria: unique *unstable* steady states (empty circle in the centre) and *stable* equilibria (2 filled circles), inducing either $\mathbb{P}_a$- or $\mathbb{P}_t$-state (figure 3*b*). $|W_S|$ and existence of $W_S$ are dependent of the combination of other parameters. In this case, the cancer cell may take either anti-apoptosis or apoptosis based on the early intracellular states. In a reverse direction, starting from $\mathbb{P}_a$-status, the system maintains the IFN-β-mediated apoptosis up to the critical bifurcation position where the STAT1 level is pulled down to the low arm with $\mathbb{P}_t$-mode as $S$ is decreased. This analysis illustrates that

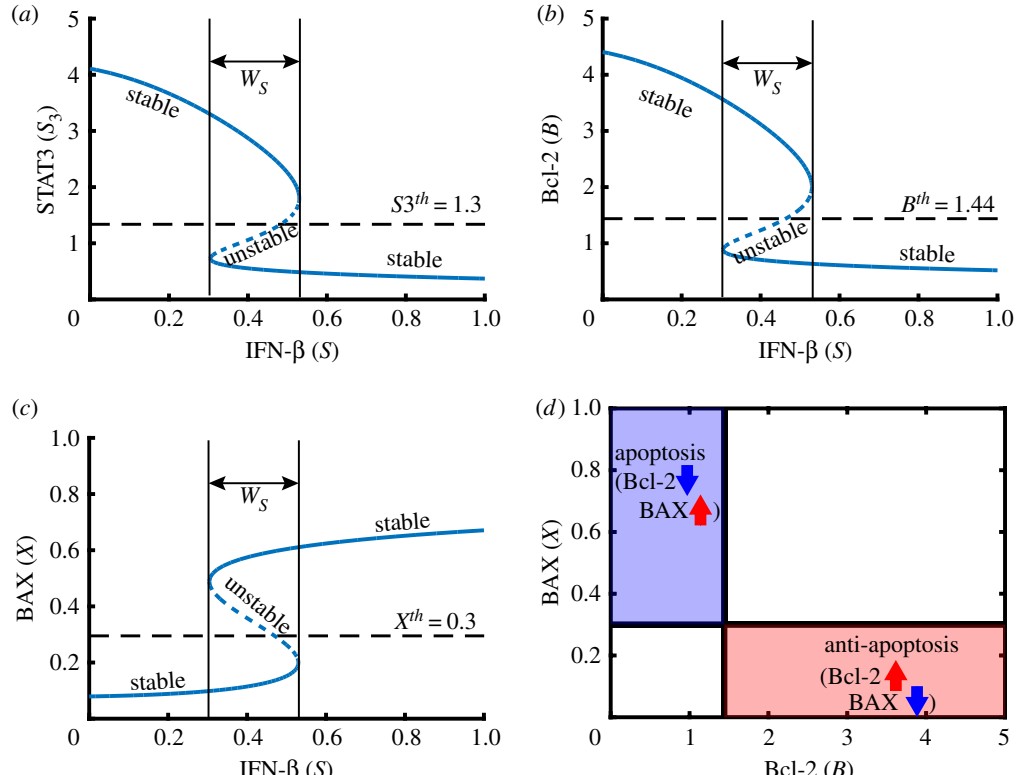

**Figure 4.** Bifurcation diagram and characterization of apoptosis when $J = 0$. (a–c) Bifurcation curves for steady states of STAT3 ($S_3$ in (a)), Bcl-2 ($B$ in (b)) and BAX ($X$ in (c)): IFN-β signals ($S$) provide an on-off switch of STAT3, Bcl-2 and BAX, induing binary modes: malignant and benign progression. $W_S = [S^m, S^M]$ = a window of bi–$stabilty$. (d) Schematic of anti-apoptosis ($\mathbb{P}_t$) and apoptosis ($\mathbb{P}_a$) regions in the $B - X$ plane. Parameters: $J = 0$. Other parameters are given in table 1.

IFN-β can be a key bifurcation parameter of the cellular response and the IFN-β-mediated cancer cell killing can be quite effective but in a nonlinear fashion.

Figure 5 shows the graphs $S_1 = S_1(S)$, $S_3 = S_3(S)$, $B = B(S)$ and $X = X(S)$ in the presence of JAK2 ($J = 1$). The levels of STAT1 and BAX stay below the threshold values ($S_1^{th}$, $X^{th}$, respectively) regardless of IFN-β stimuli while STAT3 and Bcl-2 levels are upregulated regardless of IFN-β signal strength. This results in uniform ($\mathbb{P}_t$)-state in the cancer cells although the level of IFN-β is high. Therefore, in the presence of JAK2, the intracellular module transits from the IFN-β-dependent apoptosis system to IFN-β-independent $\mathbb{P}_t$-state, implying the necessity of adjuvant therapy such as JAK2 inhibitor (DDP) in addition to the conventional IFN-β treatment.

In order to see how sensitive the concentrations of main variables (STAT1, STAT3, Bcl-2, BAX, IFN-β, JAK2, DDP and tumour) are to 26 parameters in the model (equations (2.9)–(2.16)) at various time points, we have performed sensitivity analysis. A partial rank correlation coefficient determines whether an increase (or decrease) in the parameter value will either decrease or increase the tumour volume and concentrations of main variables at a given time. See electronic supplementary material for more details.

## 3.2. Therapeutic approaches: apoptotic cell death by IFN-β and DDP

We study the therapeutic effect of IFN-β and DDP on slowing down the tumour. It is assumed that the tumour is treated with IFN-β and DDP on $[t_i, t_i + h_s]$, $i = 1, ..., N_S$ with $\tau_s (= t_{i+1} - t_i)$, $i = 1, ..., N_S - 1$ and $[t_j, t_j + h_d]$, $j = 1, ..., N_D$ with $\tau_d (= t_{j+1} - t_j)$, $j = 1, ..., N_D - 1$, respectively. Here, $N_S$, $N_D$ are the total number of infusion of IFN-β and DDP, respectively. In order to take into account the time-dependent injection of IFN-β and DDP as in clinic, we use the following ODEs of IFN-β and DDP concentrations

$$\frac{dS}{dt} = \sum_{i=1}^{N_S} u_S I_{[t_i, t_i + h_s]} - \mu_S S, \tag{3.3}$$

$$\frac{dD}{dt} = \sum_{j=1}^{N_D} u_D I_{[t_j, t_j + h_d]} - \mu_D D, \tag{3.4}$$

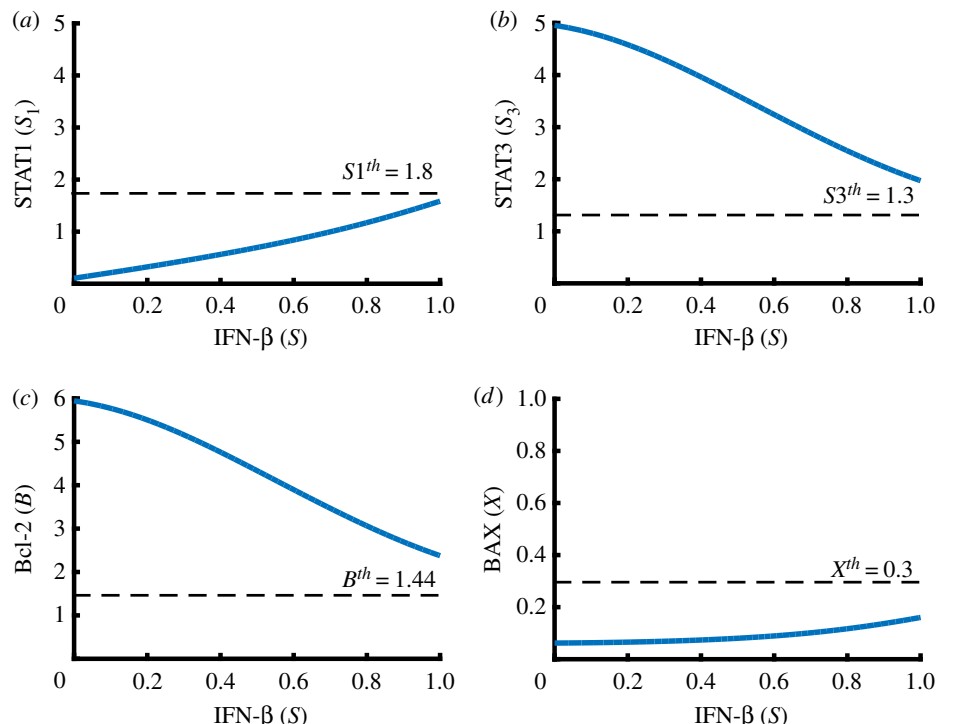

**Figure 5.** Intracellular response to IFN-β when $J > 0$. When $J > 0$, IFN-β induces uniform responses in intracellular states: downregulation of STAT1/BAX (i.e. $S_1 < S_1^{th}$, $X < X^{th}$, $\forall S^\sim (0 \leq S \leq 1)$) and upregulation of STAT3/Bcl-2 (i.e. $S_3 > S_3^{th}$, $B > B^{th}$, $\forall S^\sim (0 \leq S \leq 1)$). $Y$-axis = steady state (SS) of given variable. Parameters: $J = 1$. Other parameters are given in table 1.

where

$$I(t) = \begin{cases} 1 & \text{if } t \in [t_i, t_i + h_s], \\ 0 & \text{otherwise.} \end{cases} \tag{3.5}$$

Time evolutions of activities of STAT1 ($S_1$), STAT3 ($S_3$), Bcl-2 ($B$) and BAX ($X$) with IFN-β infusion at $t = 1$, 3, 5, 7, 9 [80] are shown in figure 6a. Here, we set $u_S = 15$. Blue dashed curves in figure 6b,c show the corresponding trajectories of the solution ($S_1(t)$, $S_3(t)$) in the $S_1 - S_3$ plane and ($B(t)$, $X(t)$) in a $B - X$ domain, respectively. Beginning from a position in the $\mathbb{P}_t$-mode (black arrows in figure 6b,c), cancer cells rapidly show a periodic response to the IFN-β and the initial $\mathbb{P}_t$-state slowly transits to the $\mathbb{P}_a$-state (blue box in figure 6b,c) by downregulated STAT3/Bcl-2 and over-expressed STAT1/BAX, maintaining the cell killing state up to about $t = 20$. Note, however, that the system converges back to the $\mathbb{P}_t$-state after $t = 20$ since the anti-cancer effect of IFN-β is creased at a later time due to early injection of IFN-β. Note also that the system stays in the $\mathbb{P}_t$-state without any transition to $\mathbb{P}_a$-mode when $u_S = 0$ (red solid curves in figures 6b,c). The corresponding changes in tumour volume (mm³) when $u_S = 0$ and $u_S = 15$ are shown figure 6d. It shows the effective anti-tumour efficacy of IFN-β even though the system returned back to the $\mathbb{P}_t$-phase after the massive tumour cell killing. Theoretical predictions well reproduce empirical data in control (circular marks; figure 6d) and IFN-β-treated cases (triangles; figure 6d) [80]. The dose response pattern at $t = 25$ (figure 6e) implies that the tumour volume is significantly decreased when the IFN-β dose is about 15, and marginally decreased beyond this dose level. Therefore, given periodic injection schedule, the model can predict the minimum dose of IFN-β for suppression of tumour growth. Figure 6f shows expression values of all intracellular variables at $t = 25$ for various IFN-β amounts (0 (PBS), 10,15,100). This illustrates a significant increase in the apoptotic agents (STAT1, BAX in figure 6f) and dramatic decrease in activities of anti-apoptotic gate keepers (STAT3, Bcl-2) as the IFN-β level increases.

We now consider a periodic infusion of both IFN-β and DDP over the time interval [0, 30] with $N_S = 3$, $\sim N_D = 3$ and various schedules. IFN-β and DDP are marked with '**S**' and '**D**', respectively, in figure 7. For example, '**SSSDDD**' indicates three consecutive injections of IFN-β followed by another three consecutive injections of DDP. In figure 7a, the relative tumour size at $t = 30$ is shown for various schedules with continuous infusion strategy with $u_S = 8.2$, $u_D = 46$. Figure 7b,c shows the time courses

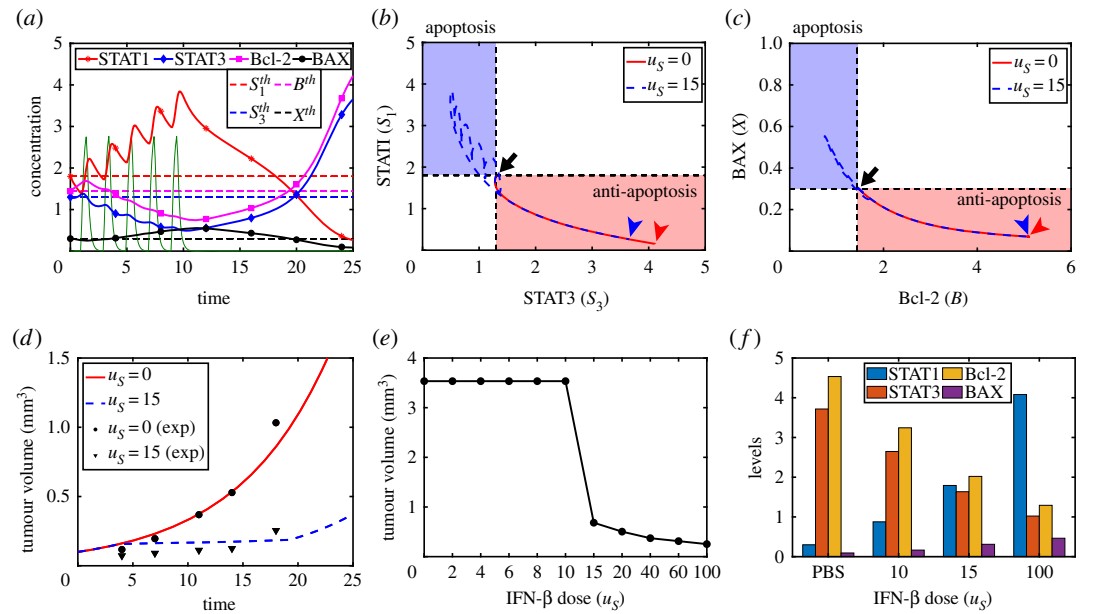

**Figure 6.** Intracellular apoptosis dynamics (STAT1, STAT3, Bcl-2, BAX) in response to IFN-β and therapeutic effect. (*a*) Time evolution of STAT1 ($S_1$), STAT3 ($S_3$), Bcl-2 ($B$) and BAX ($X$) after IFN-β treatment (green) at $t = 1, 3, 5, 7, 9$ with $u_S = 15$. (*b,c*) Solution trajectories of the intracellular variables (STAT1-STAT3 in (*b*) and Bcl-2-BAX in (*c*)) in the $S_1 - S_3$ and $B - X$ plane, respectively, with two IFN-β doses: $u_S = 0$ (red solid), $u_S = 15$ (blue dotted; corresponding to (*a*)). Arrow = initial condition, red arrow head = position of solutions with $u_S = 0$ at final time. blue arrow head = position of solutions with $u_S = 15$ at final time. (*d*) Comparison with experimental data: growth patterns of a tumour in control (simulation; solid red curves), and IFN-β-treatment (dashed curves) from our model, control case from experimental (PBS; filled circles), and IFN-β injection case from experimental (filled triangles). (*e*) Dose response curve when $u_S = 0, 2, 4, 6, 8, 10, 15, 20, 40, 60, 100$. (*f*) Average field of STAT1 (blue), STAT3 (red), Bcl-2 (yellow) and BAX (purple) when $u_S = 0$ (PBS), 10, 15, 100.

of levels of STAT1 ($S_1$), STAT3 ($S_3$), Bcl-2 ($B$) and BAX ($X$) in response to (**SSSDDD**) and (**DSDDSS**), respectively, among various cases in figure 7*a*. The (**SSSDDD**) strategy leads to the best outcome in reducing tumor volume (red solid curve in figure 7*d*) by successfully suppressing Bcl-2 for a long time despite increasing activities of the anti-apoptotic Bcl-2 at the last (figure 7*b*). On the other hand, the (**DSDDSS**) schedule results in the worst outcome (blue dashed curve in figure 7*d*) due to frequent uprising of Bcl-2 regulator (figure 7*c*). For further analysis, we divided all cases in figure 7*a* into two groups: schedules with initial injection of IFN-β (left panel in figure 7*e*) and schedules with initial injection of DDP (right panel in figure 7*e*). For example, 'SSSDDD', 'SSDSDD', ..., 'SDDDSS' belong to the first group. The levels of these variables in each case are marked with a black star. We then calculated the average levels of STAT1, STAT3, Bcl-2, BAX and tumour volume in all cases figure 7*a* from each group. Differences in expression in Bcl-2 and BAX, two key players in the cell-death program, affect differences in tumour volume. For instance, the tumour size of the first tier is smaller than that of the second schedule due to relatively lower value of Bcl-2, anti-apoptosis factor. This result implies that the IFN-β-first strategy relative to initial DDP injection can be more effective in reducing tumor size through effective induction of apoptosis. For the second analysis in figure 7*f*, we divided all cases in figure 7*a* into two groups: schedules with weighted injection of IFN-β in the first half (left panel), and schedules with weighted injection of IFN-β in the second half (right panel). For example, 'SSDSDD', 'DSSDDS' belong to the first group while 'DDSSDS', 'DDSSSD' belong to the second group. When IFN-β is distributed with more frequency in the first half, Bcl-2 expression (3rd yellow bar on the left panel, figure 7*f*) is lower compared to the case in the second half (3rd yellow bar on the right panel, figure 7*f*), leading to the smaller tumor size (5th green bar, figure 7*f*). Therefore, the model suggests that frequent IFN-β injections early on can result in better overall anti-tumour efficacy.

In figure 7, the IFN-β dose rate ($u_S$) is fixed for all cases. In figure 8*a*, we investigate how much IFN-β is needed for reduction in the tumour size by 50% compared to the case without IFN-β. Here, injection schedule ($h_s = 5$) is fixed as in figure 7. This implies that injection costs of IFN-β would be very different under various injection orders. We get the best results in the case of 'SSSDDD' with the minimum injection rate ($u_S = 5.2$), leading to the relatively smaller total dose ($T_D = 26$) over one injection period. On

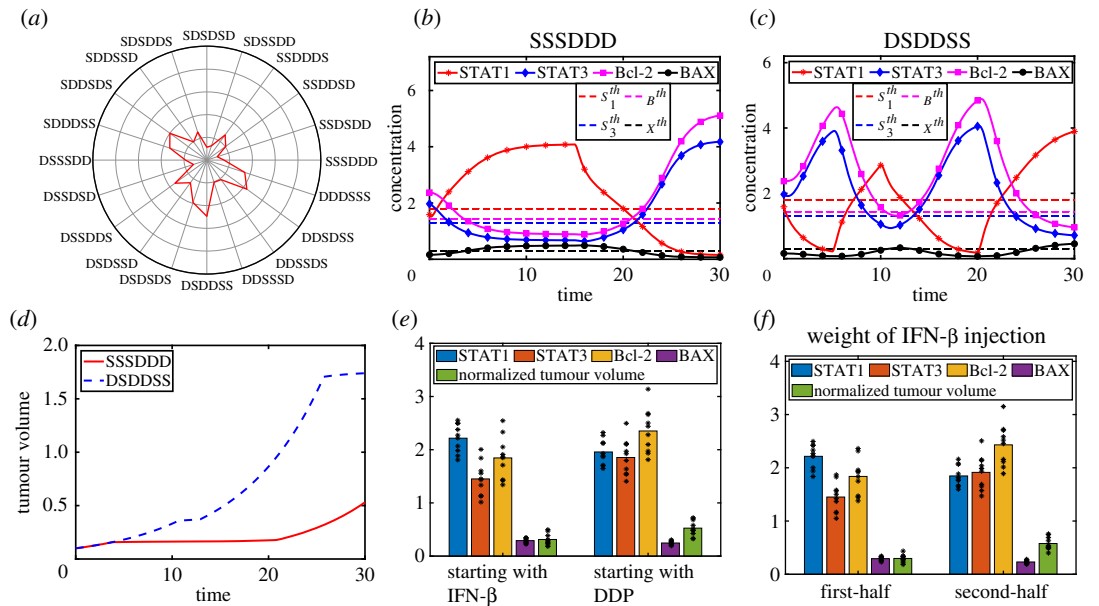

**Figure 7.** Effect of the injection order of IFN-β and DDP on tumour growth. (a) Normalized tumour volume for 20 different schedules. (b,c) Time courses of the STAT1 ($S_1$), STAT3 ($S_3$), Bcl-2 ($B$) and BAX ($X$) in response to the 'SSSDDD' (b) and 'DSDDSS' (c) cases. (d) Dynamics of the tumour size in the 'SSSDDD' (solid red) and 'DSDDSS' (dashed blue) cases. (e) Average levels of STAT1, STAT3, Bcl-2 and BAX, and tumour volume of all cases in (a) in two categories: schedules with initial injection of IFN-β (left panel), schedules with initial injection of DDP (right panel). (f) Average levels of STAT1, STAT3, Bcl-2 and BAX, and tumour volume of all cases in (a) in two categories: schedules with weighted injection of IFN-β in the first half (left panel), schedules with weighted injection of IFN-β in the second half (right panel).

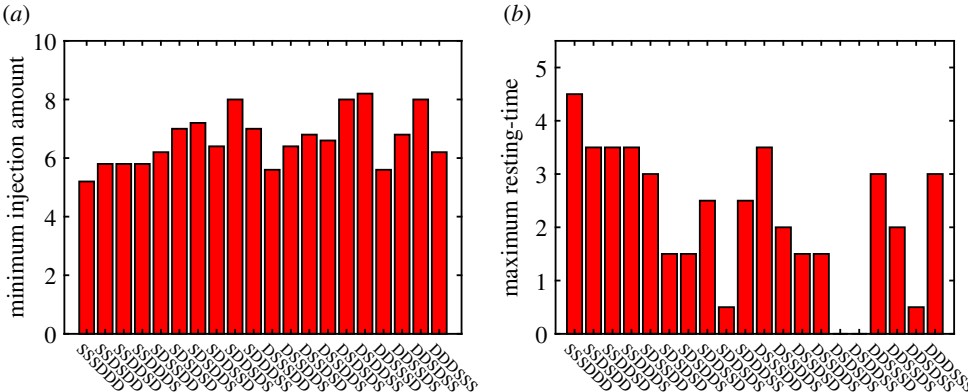

**Figure 8.** (a) The minimum injection dose of IFN-β in the permutation sequences of the two drugs in figure 7, for reduction in the tumour size by 50% compared to the case without IFN-β (control) while $h_s$ is fixed. (b) Maximum resting-time of IFN-β in the permutation sequences of the two drugs, for the same degree of tumour size reduction (50%) compared to the control.

the contrary, we get the worst case, '**DSDDSS**', with $u_S = 8.2$, resulting in the higher total dose ($T_D = 41$) over the same period. As a result, one would need at least 1.5-fold drugs in the worst case compared to the best scenario. In figure 8b, we calculate maximum resting-time of IFN-β in the permutation sequences of the two drugs, for the same degree of tumour size reduction (50%) compared to the control while $u_S$ is fixed. In the best case (**SSSDDD**), the maximum resting-time is 4.5 (i.e. $h_s = 0.5$). On the contrary, IFN-β has to be injected constantly without resting-time in the worst cases (**DSDSDS**, **DSDDSS**). Therefore, we can set a treatment schedule that provides maximum resting-time with relatively low IFN-β dose.

Figure 9 shows the tumour size in response to the combination therapy (IFN-β+DDP) with various duration of IFN-β injection ($h_s = 2, 1, 0.5, 0.1$). Here, the total amount of IFN-β and DDP was fixed: $\int_{t_s}^{t_e} u_S \, \mathrm{d}t = 123$ and $\int_{t_s}^{t_e} u_D \, \mathrm{d}t = 690$, respectively; $t_s = 0$, $t_e = 30$. The tumour size was normalized relative to the tumour volume in the case without IFN-β treatment. As $h_s$ is decreased ($A \rightarrow B \rightarrow C \rightarrow D$), the tumour size is increased overall. However, the combination therapy in some cases with low $h_s$'s

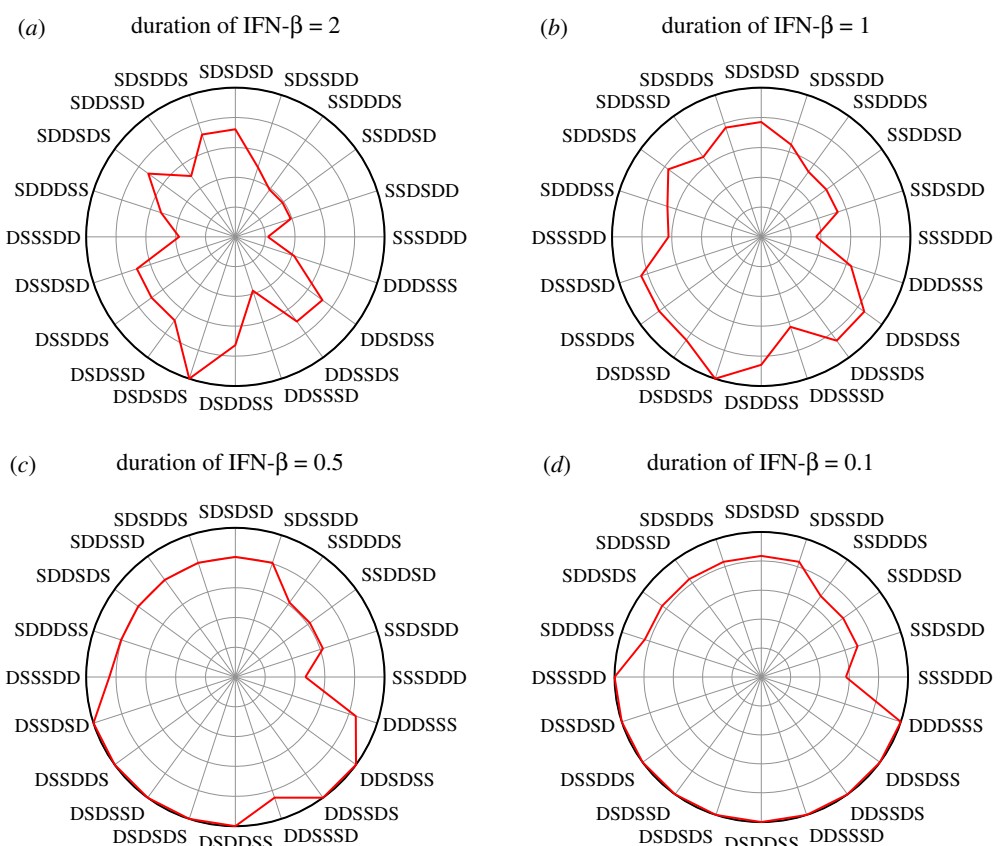

**Figure 9.** Effect of duration of IFN-β injection on tumour size. The normalized tumour volume with various duration of IFN-β injection: (*a*) $h_S = 2$, (*b*) $h_S = 1$, (*c*) $h_S = 0.5$ and (*d*) $h_S = 0.1$.

(figure 9*d*) was not effective in reducing the tumour size. For example, the tumour size at final time was same as the case without IFN-β treatment in the cases like (**DSSSDD, DSSDSD, DDDSSS**).

These results essentially suggest the optimally controlled treatment option for avoiding side effects and minimizing administrative costs at clinics.

## 3.3. Optimal control approach

Here, we consider two weight parameter sets of an objective function in an optimal control formulation. The equation of objective function, equation (2.17), has eight parameters ($A_1$, $A_2$, $A_3$, $C_1$, $C_2$, $C_3$, $C_4$, $\bar{T}$). From now on, the following two strategies are considered: (i) Strategy I: control for the entire schedule (upto $t = 30$). (i) Strategy II: control for the order of injection of IFN-β and DDP [58].

Strategy I:

In figure 10, we test the anti-tumour efficacy for three different infusion cases: alternating, constant, optimal infusion methods of IFN-β (solid curve) and DDP (dashed curve). In the alternating method, IFN-β is administered with $u_S = 5.2344$ and $\tau_s = 4$ while DDP is administered with $u_D = 53.4109$ and $\tau_d = 2$ (figure 10*a*). On the other hand, we set $u_S = 4.1875$ and $u_D = 24.9251$ in the constant injection method (figure 10*b*). Figure 10*c* shows the control profiles of IFN-β (green, solid) and DDP (pink, dashed) from the optimal scheme. Here, the shaded area indicates the corresponding control amount. We set the parameter to $(A_1, A_2, A_3, C_1, C_2, C_3, C_4, \bar{T}) = (1, 0, 0, 4 \times 10^{-5}, 1 \times 10^{-5}, 4 \times 10^{-5}, 1 \times 10^{-5}, 0)$. Note that in all three cases, the total amount of both IFN-β and DDP was fixed for comparison. See figure 10*g* for the temporal profile of cumulative injection amount for three cases. Figure 10*d*,*e* shows the corresponding time courses of IFN-β and DDP levels, respectively, in three (alternating, constant, optimal) cases. The tumour size in response to the constant injection (blue; figure 10*h*) is much smaller than one in alternating injection strategy (black; figure 10*h*) due to high levels of apoptosis within the cancer cells (high BAX & low Bcl-2 in figure 10*f*). However, the optimally controlled injection schedule provides the best results, maintaining the lowest tumour volume (red; figure 10*h*) due to effective control of tumour volume early on.

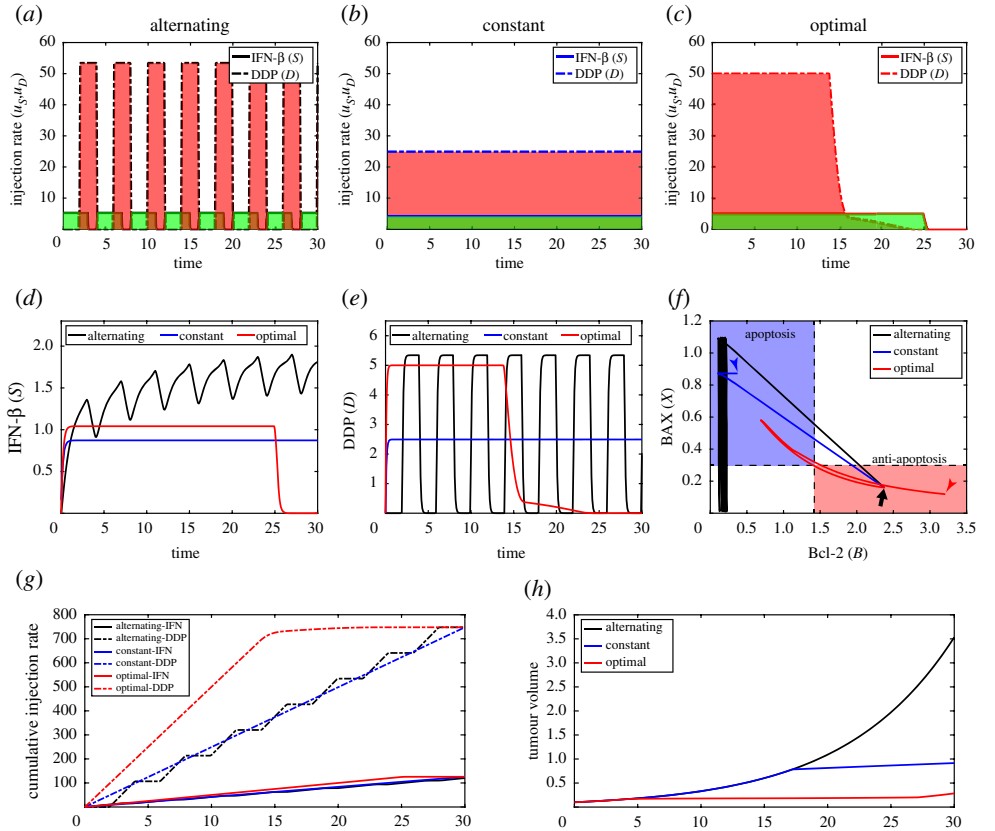

**Figure 10.** Inhibition of tumour growth by optimal control (Strategy I). (*a*) Profiles of IFN-β (green, black solid curve) and DDP (pink, black dashed curve) for alternating scheme. (*b*) Profiles of IFN-β (green, blue solid curve) and DDP (pink, blue dashed curve) for constant scheme. (*c*) Optimal control profiles of IFN-β (green, solid) and DDP (pink, dashed). (*d,e*) Time courses of IFN-β (*d*) and DDP (*e*) corresponding to three cases in (*a–c*). (*f*) Flow of solutions (Bcl-2, BAX) in the phase plane. (*g,h*) Time evolution of the cumulative injection rates (*g*) and corresponding tumour size (*h*) in three cases in (*a–c*).

We study how the half-life of IFN-β affects control of the tumour volume by changing the parameter $\mu_S (= 5.5, 4.8,$ and $3.2)$. The baseline value of the IFN-β's half-life is 5 h corresponding to the decay rate value $\mu_S = 4.8$. As the half-life is increased (i.e. when $\mu_S$ is decreased), the final tumour size is decreased in all three cases (figure 11). The tumour volume at final time is very sensitive to the changes in $\mu_S$ in both alternative (blue bar) and constant (yellow bar) injection strategies. In the optimal control case, while the tumour volume is still decreased as $\mu_S$ is decreased, the overall sensitivity is much lower (red bar). Therefore, this result suggest that the optimal control strategy can be very effective relative to the other two strategies in reducing the tumour size regardless of fluctuations in IFN-β supply.

Strategy II:

We now analyse the dynamical changes when either duration or amount of IFN-β is fixed. Here, we consider optimal control strategies of determining the optimal duration and amount in two different alternating schemes of IFN-β and DDP in figure 12. We set the parameters to $(A_1, A_2, A_3, C_1, C_2, C_3, C_4, \bar{T}) = (12, 1, 4, 2 \times 10^{-2}, 0, 2 \times 10^{-2}, 0, 1.7634)$. We first consider the best scenario (**SSSDDD**) in figure 8 and apply the optimal control frame. Optimally controlled multiple injection of IFN-β early on (figure 12*a*; blue solid curve in figure 12*c*) can effectively initiate the cell-death program in tumour cells (figure 12*d*), leading to the well controlled tumour volume (blue solid; figure 12*f*). On the other hand, the temporal profile of IFN-β in the optimal control framework in the worst schedule ('**DSDDSS**') in figure 8 is shown in figure 12*b*. In this case, more dynamical changes in the intracellular variables are observed (figure 12*e*) and temporal profile of tumour volume reduction (%) (figure 12*e*) is very different from the (**SSSDDD**) scheme. Note that the infusion patterns of IFN-β, $u_S$, in the '**SSSDDD**' and '**DSDDSS**' schemes are different (figure 12*a,b*). In the '**SSSDDD**' case (figure 12*a*), the JAK2 level is high ($J \approx 1$) due to lack of DDP infusion in the first sequence and higher initial doses of IFN-β is required to induce the apoptotic state of cancer cells (figure 5). On the contrary, in the '**DSDDSS**' case (figure 12*b*), a high dose of IFN-β is not necessary

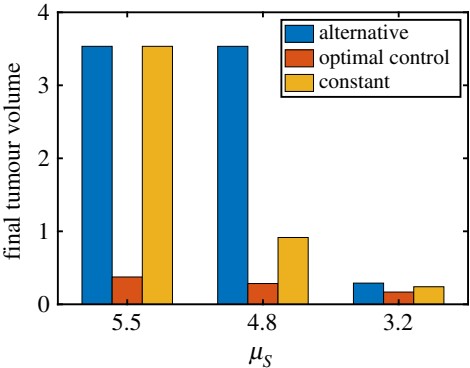

**Figure 11.** Effect of the half-life of IFN-β on different anti-cancer strategies. Final tumour size in alternative, constant and optimal control strategies with various decay rates of IFN-β ($\mu_S = 5.5$, $4.8^*$, $3.2$). See other parameters in table 1.

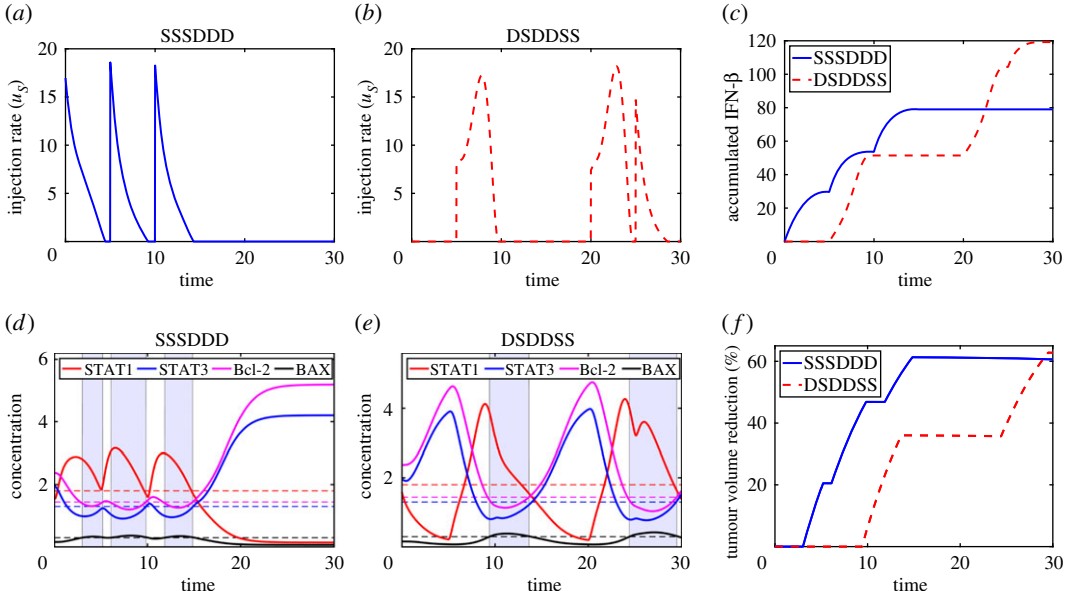

**Figure 12.** Dynamics of the system corresponding to two alternating schemes of IFN-β and DDP with optimal control. (*a,b*) Temporal changes in IFN-β infusion in the '**SSSDDD**' and the '**DSDDSS**' scheme, respectively. (*c*) Time courses of the accumulated IFN-β in two cases. (*d,e*) Time courses of the intracellular molecules (STAT1 ($S_1$), STAT3 ($S_3$), Bcl-2 ($B$) and BAX ($X$)) in the '**SSSDDD**' and '**DSDDSS**' scheme, respectively. The apoptotic state of cancer cells is marked in blue boxes. (*f*) Reduced tumour sizes of two schemes relative to the base case.

due to low JAK2 level ($J \approx 0$) from DDP infusion in the beginning. In other words, until the moment the JAK2 level lowered by the DDP is raised again, a relatively low dose of IFN-β is enough to induce the apoptotic status of cancer cells (figure 4). The minimum amount of IFN-β for 50% reduction in tumour volume is 78 in '**SSSDDD**' and 123 in '**DSDDSS**' without the optimal control (figure 8). While the accumulated amount of IFN-β with the optimal control scheme (79.01 in '**SSSDDD**' and 119.2 in '**DSDDSS**' (figure 12*c*)) is similar to the original scheme in figure 8, the tumour volume was decreased by 60.6% and 62.8% relative to the control case in the '**SSSDDD**' and '**DSDDSS**', respectively (figure 12*f*). Overall, this illustrates that the optimal control scheme can be effective in reducing the tumour size as well as costs.

## 4. Conclusion

In this work, we adapted mathematical modelling to investigate the role of regulatory cytokines (IFN-β, JAK2) on tumour growth through the corresponding intracellular signalling networks and mutual inhibition mechanism between STAT1 and STAT3. Careful examination of the model revealed the

critical, dynamical transition between apoptotic and anti-apoptotic status of cancer cells in response to fluctuating IFN-β (figure 3). Bi-stable regulation of the STAT1 and STAT3 can induce a phenotypic onset that promotes or suppresses cancer progression (figures 4 and 5). While IFNs have been proposed as the only treatment option available for prevention of recurrence and overall survival [107], careful consideration of high dose IFNs administration in the IIb/III stage is needed [108]. Therefore, despite extensive clinical application, it is essential to design the optimally controlled schedule of IFNs including doses and infusion periods [108]. In this study, we seek to find the optimal infusion scheme of a combined therapy (IFN-β+DDP) by optimal control theory.

There are various factors that may induce apoptosis in lung cancer [29–31,35]. In this study, we focused on the apoptosis process using the JAK-STAT signalling pathway. Due to a positive correlation between STAT3 and the survival of tumour cells in response to cisplatin, JAK-STAT signalling, which induces apoptosis through IFNs and cisplatin, is an important pathway for cancer cell killing [109]. We first investigated the effect of the combination (IFN-β+DDP) therapy with various alternating sequences on tumour growth by monitoring the anti-apoptotic or apoptotic status of cancer cells (figures 7–9). We found that these various injection sequences can generate very different clinical outcomes and that the IFN-β injection cycle in each combination panel induces a greater effect on suppressing tumour growth (figures 7 and 12). Initial IFN-β dose can be determined based on whether DDP was injected or not just before IFN-β (figures 4, 5, 12), suggesting the importance of infusion sequence. Thus, we developed the optimal injection strategies of IFN-β and DDP when the total dose is fixed (figures 10 and 11). The goal of the optimal strategies is to maximize the anti-tumour efficacy with minimal side effects [110–112] by control infusion protocols of both IFN-β and DDP (figure 10). In this study, the model predicted that the optimally controlled schedule of those two drugs may provide better anti-tumour efficacy with minimal costs. The mathematical model in this study may provide a comprehensive understanding of the IFN-β/JAK-induced STAT signalling network and the associated optimal control method may suggest an optimal infusion strategy of anti-cancer drugs in clinical setting.

Conventional anti-cancer agents target an apoptosis pathway in cancer cells. However, a combination treatment may result in unexpected results in signalling network. For example, while bortezomib, an anti-cancer agent, induces tumour cell killing by apoptotic pathway, the bortezomib treatment combined with oncolytic viruses can induce a more critical death program called *necroptosis* in cancer cells, causing synergistic anti-tumour effect [6,7,113]. Anti-tumour efficacy can be even better when an adjuvant therapy by NK cells is added to a combination therapy or when NK cells are completely removed from tumour microenvironment [7]. Therefore, a combination of two anti-cancer drugs may not always result in better results due to the complexity of the biological system [7]. Therefore, optimal control of drug infusions in a combination therapy [58] would be useful in assessing injection schedules of various anti-cancer drugs while minimizing several costs.

Our study has three main limitations:

(i) Delivery of anti-cancer drugs is a complex process including transport of the anti-cancer agent through tissue. In this work, we did not take into account spatial movement of these agents. Therefore, a more general framework such as partial differential equations (PDEs) instead of the ODE model in this work may better describe the spatial transport of drugs. However, development of an optimal control scheme in the PDE model for a larger multi-scale system including blood vessels is still a challenge and we plan to investigate the spatial aspect of drug transport.

(ii) Our optimal control problems have not considered a linear form of control which may be more realistic than a quadratic form. In particular, $\int_{t_s}^{t_e} u_S(t)dt$ and $\int_{t_s}^{t_e} u_D(t)\,dt$ represent the total amount of IFN-β and DDP, respectively. Therefore, this type of control forms in a minimization problem can be clearly interpreted as drug toxicity or cost by introducing weights [105,106,114]. We plan to develop a linear form of controls in a feasible setting of tumour models in future work.

(iii) Tumour microenvironment and signalling networks play a major role in cancer progression and invasion. We did not take into account various microenvironmental factors including immune cells (macrophages, neutrophils, T cells, Th cells, T regs and NK cells), cytokines/chemokines, and inter- and intra-cellular molecules. We plan to study these critical factors in future work.

Our mathematical formulation in this work can contribute to development of a new theoretical approach for other cell killing mechanisms such as necroptosis [115–117] and autophagy [118] in cancer by the

optimal control of key regulators in an ODE/PDE [6–8,58,119] or multi-scale hybrid model [5,9,28,32,120–123] where key cellular death programs within the cancer cells can be taken into account at individual cell level in the multi-scale system.

Data accessibility. This article has no additional data.
Authors' contributions. Formal analysis: D.L., J.L. Investigation: D.L., J.L. Methodology: J.L., Y.K. Supervision: Y.K. Validation: Y.K. Visualization: D.L. Writing-original draft: D.L. Writing-review and editing: Y.K.
Competing interests. We declare we have no competing interests.
Funding. This work was supported by the National Research Foundation of Korea (NRF) grant funded by the Korea government (MSIT) (no. 2021R1A2C1010891) (Y.K.) and Konkuk University's research support program for its faculty on sabbatical leave in 2020 (Y.K.).

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
