## [Peer Review File · Royal Society Open Science]

Review History

RSOS-210594.R0 (Original submission)

Review form: Reviewer 1

Is the manuscript scientifically sound in its present form?

Yes

Are the interpretations and conclusions justified by the results?

Yes

Is the language acceptable?

Yes

Do you have any ethical concerns with this paper?

No

Have you any concerns about statistical analyses in this paper?

No

Recommendation?

Major revision is needed (please make suggestions in comments)

Comments to the Author(s)

The authors integrated a tumor growth model with intracellular signaling model. The model generated a bi-stability condition for cell state - apoptosis or anti-apoptosis. They considered two types of intramuscular injections (IFN-beta and cisplatin) as a medication to modulate the cell signal, which in turn controls the tumor growth. An optimal control theory was used to deduce optimal schedule to inject to reduce tumor growth more than constant and alternating treatment. I believe the findings of this study is a useful contribution to mathematical oncology field. A more detailed explanation of mathematical analysis and parameter estimation will be beneficial to readers.

Major Issues:

1. Model parameters are mentioned as estimated in the table and a detailed information is presented as supplementary. A brief explanation of sensitivity analysis and parameter estimation process in the main text will be useful for readers.
2. Mathematical details of the equilibria and bifurcation analysis would be helpful for readers to follow this study
3. How authors derived optimal control objective function equation (15)? What kind of numerical solution technique was used to solve the equation?
4. Between strategy 1 and 2, which one is better in what condition?
5. Discussion of implication, merit, a couple of limitation, future outlook would be helpful.

Minor issues:

1. The model is developed in equation (1) to (4) and then non-dimensionalized in (7) to (10). Further, the model is extended incorporating equations (11) to (14) without discussing the dimensional relevance of the variables and parameters appeared in (11) to (14) with the parameters in equation (6).
2. Equation (14): " $u_D \leftarrow u_D(t)$ ".
3. Page 5, line 57 to Page 6, line 53: "Weight for each control is provided by parameters C_1, C_2, C_3, C_4 . Linear ($u_S(t)$) and quadratic ($u_S(t)^2$) forms in Eq. (15) represent the costs". Here the weights " C_1, C_2, C_3, C_4 " are associated with costs, and " $u_S(t), u_D(t)$ " should be associated with the doses of injection. Further, as " C_1, C_3 " are weight parameters for different degree of the same control, should they be necessarily different?
4. T^* is mentioned in the table but not appeared in the text.
5. Please provide the figure captions in detail. For example, the meaning of the light red and light blue shaded region in Fig 3(A,B,C) is not mentioned in the caption. Further, in Fig 4D, the shaded regions are dark. I wonder if the all the shaded regions are same or light or dark shades have different meaning.
6. Page 10, line 21: Please check "decreased activities of STAT1(A? increased?)/BAX (D? increased) and increased (decreased) levels of STAT3/Bcl-2".
7. Page 10, line 33: "Here, NS, ND are the total infusion of IFN- β " \leftarrow "Here, NS, ND are the total number of infusion of IFN- β ".
8. Page 11, line 50: "tumor size at $t=30$ days is shown", t is non-dimensional time, but here it is mentioned in a unit of day.
9. Page 11, 12, 13: Several " " are converted to " \ " .
10. Page 14, line 45: "following two cases are considered." \leftarrow "following two strategies are considered."
11. Page 14, line 45: "Strategy I: control for the entire schedule of 30 days ($T = 30$)". Here, time is denoted by T , which is the symbol for tumor volume.
12. Page 14, line 54,55: "(1, 0, 0, $4d^{-5}, 1d^{-5}, 4d^{-5}, 1d^{-5}, 0$)". Does the " $1d^{-5}$ " means " $1e^{-5}$ " here?

Review form: Reviewer 2

Is the manuscript scientifically sound in its present form?

Yes

Are the interpretations and conclusions justified by the results?

Yes

Is the language acceptable?

Yes

Do you have any ethical concerns with this paper?

No

Have you any concerns about statistical analyses in this paper?

No

Recommendation?

Accept with minor revision (please list in comments)

Comments to the Author(s)

This paper addresses intracellular signal transduction systems, including the STAT family. In particular, the control of the important bistable structures of cancer's apoptotic and anti-apoptotic states is investigated. This paper is the first to study the mechanism of apoptosis via the JAK-STAT signaling pathway using optimal control theory. As a therapeutic application, this study aims to find the optimal injection scheme of combination therapy (IFN- β + DDP) by optimal control theory. The model predicted that an optimally controlled schedule of these two drugs could provide better antitumor effects at minimal cost. Both the methods and results are new and valuable, and I recommend accepting this article for publication.

Minor points:

1. [Page 4; Line 13; Equation (5)] Should $f_2(s)$ be $f_2(s_j)$?

Review form: Reviewer 3

Is the manuscript scientifically sound in its present form?

No

Are the interpretations and conclusions justified by the results?

Yes

Is the language acceptable?

No

Do you have any ethical concerns with this paper?

No

Have you any concerns about statistical analyses in this paper?

No

Recommendation?

Major revision is needed (please make suggestions in comments)

Comments to the Author(s)

In general the background referencing is incomplete. Researchers other than Kim et al. have done considerable work in modeling the Bcl-2 mediated regulation of apoptosis in cancer cells.

The Introduction is a bit confusing. It will help eg at the end of the first paragraph to highlight knowledge gaps that mathematical modeling can fill. What is the need or motivation for the model?

At the end of the introduction it is not clear what the aim is. What will be achieved? What is new, in terms of the model? I get no sense of what the paper really accomplishes.

So many parameters have been 'Est' (see Table 1) that one cannot really have confidence in them. Estimated from where? Are these fit to some data? Why are most of them '1'?

Abstract

Doesn't really mention what is novel about this work. Also should be mathematically more precise, eg, what is being optimized, what is the constraint?

Line 11 – what specifically is meant by 'immunity'?

Line 13 – 'anti-apoptosis' isn't really cellular fate.

Introduction

Needs extensive grammar revision. Eg, 'most fatal killer' is poor English.

Paragraph 1 – "... two different dichotomous states of cell death: (i) an apoptosis progression and (ii) an anti-apoptosis progression." This is confusing. What is meant by anti-apoptosis progression? Does this also lead to cell death? If not, then how can it be a state of cell death?

Paragraph 1 – What is meant by "apoptosis-mediated therapy"? This seems to be poorly worded.

Paragraph 2 – "However, there was no study of apoptosis mechanism through JAK-STAT signaling pathway with optimal control theory " This sentence has a couple of issues. First, it is poorly worded. Second, it is oddly specific. Is the claim that no mathematical studies of JAK-STAT mediation of apoptosis have been published? Or that no mathematical studies of JAK-STAT mediation of apoptosis using in some manner optimal control have been published? If the latter is the case, then this is hardly a novel study. In my experience, most mathematical models of cancer response to therapy include some manner of treatment optimization, whether or not they use optimal control.

Fig 1 should be referenced in paragraph 1 where the biology is introduced.

Methodology

Paragraph 1 – why is time t with an overbar? This is not standard and what is the motivation?

Where do the equations (1) – (4) come from? That is, why these functional form choices? Have they been derived from first principles? Are they purely phenomenological? How are they justified?

In general, it would be helpful to explain what the terms on the right hand side are doing rather than get into listing every parameter. Also, please include in the main text, a table with model variables and parameters and all units.

What is the difference between capital S (see paragraph 3 of introduction) and lowercase s (Page 5, lines 8-9)? What do you mean by IFN activity? This does not have any obvious mathematical meaning. Do you mean concentration of IFN? What units?

Page 5, Lines 47-48 – I do not know what you mean by “in genetic classification”. Further in this paragraph, what biological observation? From where? In what cells? Is this data published somewhere?

In equation (11) – does the proliferation become negative when the apoptosis condition is satisfied? Then why have a separate apoptosis term? If not, then why does proliferation rate depend on apoptosis threshold? Should it not have it's own threshold? Where are the biological data justifying this? Is equation (11) also non-dimensional?

Page 6, Lines 23-24 – what is meant by alternating injection, ie, alternating with what? By the way, this notation is very confusing. Stats are denoted by S_i and drug is also S?

Page 6, Lines 29-31 – Not clear at all what is meant by this: “Dose of DDP is determined on covering section in the neighborhood of injection spot [68] due to different anti-tumor efficacy in the chemotherapy”

Decision letter (RSOS-210594.R0)

Dear Dr Kim

On behalf of the Editors, we are pleased to inform you that your Manuscript RSOS-210594 "Mathematical model of STAT signaling pathways in cancer development and optimal control approaches" has been accepted for publication in Royal Society Open Science subject to minor revision in accordance with the referees' reports. Please find the referees' comments along with any feedback from the Editors below my signature.

Please submit your revised manuscript and required files (see below) no later than 7 days from today's (ie 05-Aug-2021) date. Note: the ScholarOne system will 'lock' if submission of the revision is attempted 7 or more days after the deadline. If you do not think you will be able to meet this deadline please contact the editorial office immediately.

Please note article processing charges apply to papers accepted for publication in Royal Society Open Science (<https://royalsocietypublishing.org/rsos/charges>). Charges will also apply to papers transferred to the journal from other Royal Society Publishing journals, as well as papers

submitted as part of our collaboration with the Royal Society of Chemistry (<https://royalsocietypublishing.org/rsos/chemistry>). Fee waivers are available but must be requested when you submit your revision (<https://royalsocietypublishing.org/rsos/waivers>).

on behalf of Professor Takashi Suzuki (Associate Editor) and Mark Chaplain (Subject Editor)
openscience@royalsociety.org

Associate Editor Comments to Author (Professor Takashi Suzuki):

Your paper is to be recommended for publications. Several comments of the referees, however, are very important and useful to make it improved. Please take regards.

Reviewer comments to Author:

Reviewer: 1

Comments to the Author(s)

The authors integrated a tumor growth model with intracellular signaling model. The model generated a bi-stability condition for cell state - apoptosis or anti-apoptosis. They considered two types of intramuscular injections (IFN-beta and cisplatin) as a medication to modulate the cell signal, which in turn controls the tumor growth. An optimal control theory was used to deduce optimal schedule to inject to reduce tumor growth more than constant and alternating treatment. I believe the findings of this study is a useful contribution to mathematical oncology field. A more detailed explanation of mathematical analysis and parameter estimation will be beneficial to readers.

Major Issues:

1. Model parameters are mentioned as estimated in the table and a detailed information is presented as supplementary. A brief explanation of sensitivity analysis and parameter estimation process in the main text will be useful for readers.
2. Mathematical details of the equilibria and bifurcation analysis would be helpful for readers to follow this study
3. How authors derived optimal control objective function equation (15)? What kind of numerical solution technique was used to solve the equation?
4. Between strategy 1 and 2, which one is better in what condition?
5. Discussion of implication, merit, a couple of limitation, future outlook would be helpful.

Minor issues:

1. The model is developed in equation (1) to (4) and then non-dimensionalized in (7) to (10). Further, the model is extended incorporating equations (11) to (14) without discussing the dimensional relevance of the variables and parameters appeared in (11) to (14) with the parameters in equation (6).
2. Equation (14): " u_D " \leftarrow " $u_D(t)$ ".
3. Page 5, line 57 to Page 6, line 53: "Weight for each control is provided by parameters C_1, C_2, C_3, C_4 . Linear ($u_S(t)$) and quadratic ($u_S(t)^2$) forms in Eq. (15) represent the costs". Here the weights " C_1, C_2, C_3, C_4 " are associated with costs, and " $u_S(t), u_D(t)$ " should be associated with the

doses of injection. Further, as “C1,C3” are weight parameters for different degree of the same control, should they be necessarily different?

4. T^* is mentioned in the table but not appeared in the text.

5. Please provide the figure captions in detail. For example, the meaning of the light red and light blue shaded region in Fig 3(A,B,C) is not mentioned in the caption. Further, in Fig 4D, the shaded regions are dark. I wonder if the all the shaded regions are same or light or dark shades have different meaning.

6. Page 10, line 21: Please check “decreased activities of STAT1(A? increased?)/BAX (D? increased) and increased (decreased) levels of STAT3/Bcl-2”.

7. Page 10, line 33: “Here, NS, ND are the total infusion of IFN- β ” ← “Here, NS, ND are the total number of infusion of IFN- β ”.

8. Page 11, line 50: “tumor size at $t = 30$ days is shown”, t is non-dimensional time, but here it is mentioned in a unit of day.

9. Page 11, 12, 13: Several “ ” are converted to “ \ ”.

10. Page 14, line 45: “following two cases are considered:” ← “following two strategies are considered:”.

11. Page 14, line 45: “Strategy I: control for the entire schedule of 30 days ($T = 30$)”. Here, time is denoted by T , which is the symbol for tumor volume.

12. Page 14, line 54,55: “(1, 0, 0, 4d-5, 1d-5, 4d-5, 1d-5, 0)”. Does the “1d-5” means “1e-5” here?

Reviewer: 2

Comments to the Author(s)

This paper addresses intracellular signal transduction systems, including the STAT family. In particular, the control of the important bistable structures of cancer's apoptotic and anti-apoptotic states is investigated. This paper is the first to study the mechanism of apoptosis via the JAK-STAT signaling pathway using optimal control theory. As a therapeutic application, this study aims to find the optimal injection scheme of combination therapy (IFN- β + DDP) by optimal control theory. The model predicted that an optimally controlled schedule of these two drugs could provide better antitumor effects at minimal cost. Both the methods and results are new and valuable, and I recommend accepting this article for publication.

Minor points:

1. [Page 4; Line 13; Equation (5)] Should $f_2(s)$ be $f_2(s,j)$?

Reviewer: 3

Comments to the Author(s)

In general the background referencing is incomplete. Researchers other than Kim et al. have done considerable work in modeling the Bcl-2 mediated regulation of apoptosis in cancer cells.

The Introduction is a bit confusing. It will help eg at the end of the first paragraph to highlight knowledge gaps that mathematical modeling can fill. What is the need or motivation for the model?

At the end of the introduction it is not clear what the aim is. What will be achieved? What is new, in terms of the model? I get no sense of what the paper really accomplishes.

So many parameters have been ‘Est’ (see Table 1) that one cannot really have confidence in them. Estimated from where? Are these fit to some data? Why are most of them ‘1’?

Abstract

Doesn't really mention what is novel about this work. Also should be mathematically more precise, eg, what is being optimized, what is the constraint?

Line 11 – what specifically is meant by 'immunity'?

Line 13 – 'anti-apoptosis' isn't really cellular fate.

Introduction

Needs extensive grammar revision. Eg, 'most fatal killer' is poor English.

Paragraph 1 – "... two different dichotomous states of cell death: (i) an apoptosis progression and (ii) an anti-apoptosis progression." This is confusing. What is meant by anti-apoptosis progression? Does this also lead to cell death? If not, then how can it be a state of cell death?

Paragraph 1 – What is meant by "apoptosis-mediated therapy"? This seems to be poorly worded.

Paragraph 2 – "However, there was no study of apoptosis mechanism through JAK-STAT signaling pathway with optimal control theory " This sentence has a couple of issues. First, it is poorly worded. Second, it is oddly specific. Is the claim that no mathematical studies of JAK-STAT mediation of apoptosis have been published? Or that no mathematical studies of JAK-STAT mediation of apoptosis using in some manner optimal control have been published? If the latter is the case, then this is hardly a novel study. In my experience, most mathematical models of cancer response to therapy include some manner of treatment optimization, whether or not they use optimal control.

Fig 1 should be referenced in paragraph 1 where the biology is introduced.

Methodology

Paragraph 1 – why is time t with an overbar? This is not standard and what is the motivation?

Where do the equations (1) – (4) come from? That is, why these functional form choices? Have they been derived from first principles? Are they purely phenomenological? How are they justified?

In general, it would be helpful to explain what the terms on the right hand side are doing rather than get into listing every parameter. Also, please include in the main text, a table with model variables and parameters and all units.

What is the difference between capital S (see paragraph 3 of introduction) and lowercase s (Page 5, lines 8-9)? What do you mean by IFN activity? This does not have any obvious mathematical meaning. Do you mean concentration of IFN? What units?

Page 5, Lines 47-48 – I do not know what you mean by "in genetic classification ". Further in this paragraph, what biological observation? From where? In what cells? Is this data published somewhere?

In equation (11) – does the proliferation become negative when the apoptosis condition is satisfied? Then why have a separate apoptosis term? If not, then why does proliferation rate depend on apoptosis threshold? Should it not have its own threshold? Where are the biological data justifying this? Is equation (11) also non-dimensional?

Page 6, Lines 23-24 – what is meant by alternating injection, ie, alternating with what? By the way, this notation is very confusing. Stats are denoted by S_i and drug is also S?

Page 6, Lines 29-31 – Not clear at all what is meant by this: “Dose of DDP is determined on covering section in the neighborhood of injection spot [68] due to different anti-tumor efficacy in the chemotherapy”

===PREPARING YOUR MANUSCRIPT===

===PREPARING YOUR REVISION IN SCHOLARONE===

<https://royalsociety.org/journals/authors/author-guidelines/#supplementary-material> to include a suitable title and informative caption. An example of appropriate titling and captioning may be found at https://figshare.com/articles/Table_S2_from_Is_there_a_trade-off_between_peak_performance_and_performance_breadth_across_temperatures_for_aerobic_scops_in_teleost_fishes_/3843624.

Author's Response to Decision Letter for (RSOS-210594.R0)

See Appendix A.

Decision letter (RSOS-210594.R1)

Dear Dr Kim,

I am pleased to inform you that your manuscript entitled "Mathematical model of STAT signaling pathways in cancer development and optimal control approaches" is now accepted for publication in Royal Society Open Science.

on behalf of Professor Takashi Suzuki (Associate Editor) and Mark Chaplain (Subject Editor)
openscience@royalsociety.org

Appendix A

Mathematical model of STAT signaling pathways in cancer development and optimal control approaches

Jonggul Lee, Donggu Lee, Yangjin Kim,

RESPONSE TO REVIEWERS' COMMENTS

September 2, 2021

REFeree 1:

The authors integrated a tumor growth model with intracellular signaling model. The model generated a bi-stability condition for cell state - apoptosis or anti-apoptosis. They considered two types of intramuscular injections (IFN-beta and cisplatin) as a medication to modulate the cell signal, which in turn controls the tumor growth. An optimal control theory was used to deduce optimal schedule to inject to reduce tumor growth more than constant and alternating treatment. I believe the findings of this study is a useful contribution to mathematical oncology field. A more detailed explanation of mathematical analysis and parameter estimation will be beneficial to readers.

Major Issues:

1. Model parameters are mentioned as estimated in the table and a detailed information is presented as supplementary. A brief explanation of sensitivity analysis and parameter estimation process in the main text will be useful for readers.

(Response): Thank you for careful reading and suggestions. For more explanation of sensitivity analysis, we replaced

“Sensitivity analysis for twenty eight parameters in the model (Eqs ((7)-(14)) was provided in Supplementary Information (SI) File.” with “In order to see how sensitive are concentrations of main variables (STAT1, STAT3, Bcl-2, BAX, IFN- β , JAK2, DDP, and tumor) to twenty six parameters in the model (Eqs (9)-(16)) at various time points, we have performed sensitivity analysis. A partial rank correlation coefficient determines whether an increase (or decrease) in the parameter value will either decrease or increase the tumor volume and concentrations of main variables at a given time. See Supplementary Information File for more details.” [page 11, 1st paragraph, lines 1-6, in the (marked) revised manuscript]

For parameter estimation, we replaced “See Table I for parameter values in

Eqs. (7)-(14). Parameter estimation was provided in Supplementary Information File.” with “See Table I for parameter values in Eqs. (9)-(16) in a dimensionless form. Parameter values in the dimensional form are listed in Table S1 in the Supplementary Information File. Since our mathematical model contains many known and unknown parameter values, we provided parameter estimation in the Supplementary Information File, which is a necessary step toward fundamental and deep understanding of the dynamical process of the mathematical model. Parameter values are calculated based on empirical data such as half-life or estimated by fitting to experimental observation based on the mathematical structure of our model.” [page 6, 5th paragraph, lines 4-5 - page 7, 1st paragraph, lines 1-5, in the (marked) revised manuscript]

We also added the following parameter estimation in Supplementary Information File:

“ λ_1, λ_2 : By using the reference value of Bcl-2 ($B^* = 10nM$), and taking the signal source of the Bcl-2, $f_3 = 5.78 \times 10^{-2}nM/h$, we estimate the dimensionless value of signaling strength to the Bcl-2 complex to be $\lambda_1 = \frac{f_3}{\mu_{S_1}B^*} = 0.2$. Similarly, by taking the reference value of BAX ($X^* = 351\mu M$), and taking the signal source of BAX, $f_4 = 2.0288\mu M/h$, we estimate the dimensionless value of signaling source to be $\lambda_2 = \frac{f_4}{\mu_{S_1}X^*} = 0.2$.

λ_3, k_5 : We assume the autocatalytic rate (a_7) of Bcl-2 is same as its negative contribution ($\mu_{S_1}B^* = (2.89 \times 10^{-2}h^{-1}) \times (1.0 \times 10^{-2}\mu M) = 2.89 \times 10^{-4}\mu M/h$), leading to the dimensionless parameter value $k_5 = \frac{a_7}{\mu_{S_1}B^*} = 1.0$. By assuming that the signaling rate ($\lambda_{STAT3} \times (1.38\mu g/mL)$) from STAT3 is at least 1.2-fold larger than its negative contribution ($\mu_{S_1}B^*$) and by taking $\lambda_{STAT3} = 2.513 \times 10^{-4}\mu M \cdot mL/(h \cdot \mu g)$, we get the dimensionless parameter value $\lambda_3 = \frac{\lambda_{STAT3}S_3^*}{\mu_{S_1}B^*} = \frac{\lambda_{STAT3} \times (1.38\mu g/mL)}{(2.89 \times 10^{-2}h^{-1}) \times (1.0 \times 10^{-2}\mu M)} = 1.2$.

k_1, k_3, k_7 : We take the dimensionless parameter value $k_1 = \frac{a_1}{\mu_{S_1}S_1^*} = 4.0$ by assuming that the autocatalytic rate (a_1) of STAT1 is at least 4-fold larger than its negative contribution ($\mu_{S_1}S_1^*$) from its decay in the absence of the inhibitory pathway from the STAT3. In a similar fashion, we get the autocatalytic rate of STAT3 (k_3) and BAX (k_7), $k_3 = \frac{a_4}{\mu_{S_1}S_3^*} = 4.0$ and $k_7 = \frac{a_{10}}{\mu_{S_1}X^*} = 4.0$.

k_2, k_4, k_6, k_8 : The Hill-type coefficients a_2, a_5, a_8, a_{11} (k_2, k_4, k_6, k_8 : parameters with dimensionless form) are fixed to be unity.

k_9, k_{10}, μ_T : In the absence of apoptosis, the carrying capacity of cancer is T_0 . However, in the presence of apoptosis, the carrying capacity is $T_0 \left(1 - \frac{\mu_T}{r \left(1 - \frac{k_9 S_1^2}{k_{10}^2 + S_1^2} \right)} \right)$. Our system assumes that most cancer cells die by the

apoptosis program. We set $k_9 = 1.0$, $k_{10} = 10$, and $\mu_T = 0.1$ and the high level of STAT1 is about 4.5. Then, the steady state value of tumor volume in the presence of apoptosis is smaller than equilibrium value in the absence of apoptosis, $T_0 \left(1 - \frac{\mu_T}{r \left(1 - \frac{k_9 S_1^2}{k_{10}^2 + S_1^2} \right)} \right) \approx 0.01 \times T_0$.

The list of dimensionless parameters and their values are given in Table I in the main text. ”

[Supplementary Information File, 7th paragraph in page 1 - 4th paragraph in page 2, in the (marked) revised manuscript]

2. Mathematical details of the equilibria and bifurcation analysis would be helpful for readers to follow this study

(Response): Thank you for suggestions. We now provided the mathematical analysis of the equilibria and stability analysis in a section (‘Mathematical Analysis (equilibria and stability)’) in the Supplementary Information File (SI) as follows:

“Mathematical analysis (Equilibria and Stability)

The dynamics of the intracellular signaling system is governed by ordinary differential equations:

$$\frac{dS_1}{dt} = \lambda_{S_1} S + \frac{k_1 k_2^2}{k_2^2 + \alpha S_3^2} - S_1 = G_1(S_1, S_3, B, X), \quad (1)$$

$$\frac{dS_3}{dt} = \frac{\lambda_k + \lambda_J J}{K + \lambda_{S_2} S} + \frac{k_3 k_4^2}{k_4^2 + \beta S_1^2} - S_3 = G_2(S_1, S_3, B, X), \quad (2)$$

$$\frac{dB}{dt} = \lambda_1 + \frac{k_5 k_6^2}{k_6^2 + \gamma S_1^2} + \lambda_3 S_3 - \mu_B B = G_3(S_1, S_3, B, X), \quad (3)$$

$$\frac{dX}{dt} = \lambda_2 + \frac{k_7 k_8^2}{k_8^2 + \delta B^2} - \mu_X X = G_4(S_1, S_3, B, X). \quad (4)$$

In order to explore the equilibrium dynamics, the equilibrium state of the system must be identified first. This is done by setting Eqs. (1)-(4) to zero

and solving for the solutions S_1^E, S_3^E, B^E , and X^E as follows:

$$S_1^E = \lambda_{S_1} S + \frac{k_1 k_2^2}{k_2^2 + \alpha(S_3^E)^2}, \quad (5)$$

$$S_3^E = \frac{\lambda_k + \lambda_J J}{K + \lambda_{S_2} S} + \frac{k_3 k_4^2}{k_4^2 + \beta(S_1^E)^2}, \quad (6)$$

$$B^E = \left(\lambda_1 + \frac{k_5 k_6^2}{k_6^2 + \gamma(S_1^E)^2} + \lambda_3 S_3^E \right) \frac{1}{\mu_B}, \quad (7)$$

$$X^E = \left(\lambda_2 + \frac{k_7 k_8^2}{k_8^2 + \delta(B^E)^2} \right) \frac{1}{\mu_X}. \quad (8)$$

We can find the equilibrium point of the system (5)-(8) by nonlinear solver MATLAB as far as all parameter values are fixed. For example, we can find the equilibrium when S is 0, 0.4, and 1; when $S = 0$, $(S_1^E, S_3^E, B^E, X^E) \approx (0.1519, 4.1098, 5.0910, 0.0698)$, when $S = 1$, $(S_1^E, S_3^E, B^E, X^E) \approx (4.3176, 0.3703, 0.5794, 0.0698)$, when $S = 0.4$, there are three equilibrium points: $(S_1^E, S_3^E, B^E, X^E) \approx (3.1415, 0.5532, 0.6991, 2.8719, 3.5983, 0.0974)$, or $(S_1^E, S_3^E, B^E, X^E) \approx (1.8920, 1.0586, 1.4072, 0.2084)$.

The Jacobian matrix, \mathcal{J} , is given by:

$$\mathcal{J} = \begin{bmatrix} \frac{\partial G_1}{\partial S_1} & \frac{\partial G_1}{\partial S_3} & \frac{\partial G_1}{\partial B} & \frac{\partial G_1}{\partial X} \\ \frac{\partial G_2}{\partial S_1} & \frac{\partial G_2}{\partial S_3} & \frac{\partial G_2}{\partial B} & \frac{\partial G_2}{\partial X} \\ \frac{\partial G_3}{\partial S_1} & \frac{\partial G_3}{\partial S_3} & \frac{\partial G_3}{\partial B} & \frac{\partial G_3}{\partial X} \\ \frac{\partial G_4}{\partial S_1} & \frac{\partial G_4}{\partial S_3} & \frac{\partial G_4}{\partial B} & \frac{\partial G_4}{\partial X} \end{bmatrix} \quad (9)$$

$$= \begin{bmatrix} -1 & -\frac{2k_1 k_2^2 \alpha S_3^E}{(k_2^2 + \alpha(S_3^E)^2)^2} & 0 & 0 \\ -\frac{2k_3 k_4^2 \beta S_1^E}{(k_4^2 + \beta(S_1^E)^2)^2} & -1 & 0 & 0 \\ -\frac{2k_5 k_6^2 \gamma S_1^E}{(k_6^2 + \gamma(S_1^E)^2)^2} & \lambda_3 & -\mu_B & 0 \\ 0 & 0 & -\frac{2k_7 k_8^2 \delta B^E}{(k_8^2 + \delta(B^E)^2)^2} & -\mu_X \end{bmatrix} \quad (10)$$

Then, the characteristic polynomial is given by

$$(\mathcal{J} - \Lambda \mathbf{I}) = (\Lambda + \mu_B)(\Lambda + \mu_X) \left(\Lambda^2 + 2\Lambda + 1 - \frac{2k_1 k_2^2 \alpha S_3^E}{(k_2^2 + \alpha(S_3^E)^2)^2} \cdot \frac{2k_3 k_4^2 \beta S_1^E}{(k_4^2 + \beta(S_1^E)^2)^2} \right) = 0 \quad (11)$$

where \mathbf{I} is the identity matrix. We get two *real* eigenvalues ($\Lambda_1 = -\mu_B < 0$ and $\Lambda_2 = -\mu_X < 0$) and remaining two *real* eigenvalues, $\Lambda_{3,4} = -1 \pm \sqrt{\frac{2k_1 k_2^2 \alpha S_3^E}{(k_2^2 + \alpha(S_3^E)^2)^2} \cdot \frac{2k_3 k_4^2 \beta S_1^E}{(k_4^2 + \beta(S_1^E)^2)^2}}$. If $\frac{2k_1 k_2^2 \alpha S_3^E}{(k_2^2 + \alpha(S_3^E)^2)^2} \cdot \frac{2k_3 k_4^2 \beta S_1^E}{(k_4^2 + \beta(S_1^E)^2)^2}$ is smaller than 1, the eigenvalues $\Lambda_{3,4}$ are negative and the equilibrium is stable. On contrary, if $\frac{2k_1 k_2^2 \alpha S_3^E}{(k_2^2 + \alpha(S_3^E)^2)^2} \cdot \frac{2k_3 k_4^2 \beta S_1^E}{(k_4^2 + \beta(S_1^E)^2)^2}$ is larger than 1, then $\Lambda_3 > 0$ and the equilibrium is unstable. By substituting parameter values in the model, we get

Figure S1: **Stability analysis.** (A,B) The value of AB and Λ_3 in the spectrum of $\text{IFN-}\beta(S)$. *Gray = unstable, white = stable.

$$\mathcal{J} = \begin{bmatrix} -1 & -\frac{12S_3^E}{(1+1.5(S_3^E)^2)^2} & 0 & 0 \\ -\frac{8S_1^E}{(1+(S_1^E)^2)^2} & -1 & 0 & 0 \\ -\frac{2S_1^E}{(1+(S_1^E)^2)^2} & 1.2 & -1.2 & 0 \\ 0 & 0 & -\frac{8B^E}{(1+(B^E)^2)^2} & -5 \end{bmatrix} \quad (12)$$

By setting $A = \frac{12S_3^E}{(1+1.5(S_3^E)^2)^2}$, $B = \frac{8S_1^E}{(1+(S_1^E)^2)^2}$, $C = \frac{2S_1^E}{(1+(S_1^E)^2)^2}$, and $D = \frac{8B^E}{(1+(B^E)^2)^2}$, we get

$$\mathcal{J} = \begin{bmatrix} -1 & -A & 0 & 0 \\ -B & -1 & 0 & 0 \\ -C & 1.2 & -1.2 & 0 \\ 0 & 0 & -D & -5 \end{bmatrix} \quad (13)$$

Then, the characteristic equation is

$$(\Lambda + 1.2)(\Lambda + 5)(\Lambda^2 + 2\Lambda + 1 - AB) = 0 \quad (14)$$

When $S = 0$, $(S_1^E, S_3^E, B^E, X^E) \approx (0.1519, 4.1098, 5.0910, 0.0698)$. Thus, $A = \frac{12S_3^E}{(1+1.5(S_3^E)^2)^2} \approx 0.0711$ and $B = \frac{8S_1^E}{(1+(S_1^E)^2)^2} \approx 1.1610$ and $AB \approx 0.0825 < 1$. Therefore, when $S = 0$, the equilibrium point is stable. When $S = 1$, AB ; $A \approx 3.0568$, $B \approx 0.0895$, and $AB \approx 0.2736 < 1$. There are three equilibrium points when $S = 0.4$. We calculated AB for each case. Two steady states $(S_1^E, S_3^E, B^E, X^E) \approx (3.1415, 0.5532, 0.7965, 0.5294)$ and $(S_1^E, S_3^E, B^E, X^E) \approx (0.6991, 2.8719, 3.5983, 0.0974)$ are stable because $AB \approx 0.6633 < 1$ and $AB \approx 0.4863 < 1$, respectively. On the other hand, the third equilibrium point $(S_1^E, S_3^E, B^E, X^E) \approx (1.8920, 1.0586, 1.4072, 0.2084)$ is unstable with $AB \approx 1.2755 > 1$.

We can calculate AB and corresponding eigenvalues for continuous spectrum of $IFN-\beta$. Fig. S1A,B shows AB and Λ_3 for various $IFN-\beta$ concentrations, respectively. The gray region represents the unstable region and stable region is marked in white. The green curve in Fig. S1A① corresponds to the lower branch in Fig. 3D in the main text and the red curve in unstable region in Fig. S1A② corresponds to the middle branch in Fig. 3D in the main text, lastly the blue curve in stable region in Fig. S1A③ corresponds to the upper branch in Fig. 3D in the main text. The corresponding eigenvalues in these three regions are shown in Fig. S1B.”

[Supplementary Information File, pages 3-5, Section ‘Mathematical Analysis (equilibria and stability)’, in the (marked) revised manuscript]

In addition, in order to make it more clear, we replaced the plain bifurcation

Figure 3: Dynamics of intracellular (STAT1-STAT3-Bcl2-BAX) module in the absence of JAK2. (A-C) Solution flow of the system (7) - (10) in the S_1-S_3 set when $S=0$ (A), 0.4 (B), 1.0 (C). Filled circle = *stable* equilibrium, empty circle = *unstable* equilibrium. Blue region = up-regulation of STAT1 + down-regulation of STAT3, pink region = down-regulation of STAT1 + up-regulation of STAT3. (D) Bifurcation curve of STAT1. Y-axis= equilibrium. $W_S = [S^m, S^M]$ = a window of bi-stability.

curves with the bifurcation curves with stability information (stable lower branch; unstable middle branch; stable upper branch) in Fig 3D and Fig

Figure 4: **Bifurcation diagram and characterization of apoptosis when $J = 0$.** (A-C) Bifurcation curves for steady states of STAT3 (S_3 in (A)), Bcl-2 (B in (B)), and BAX (X in (C)): IFN- β signals (S) provide an on-off switch of STAT3, Bcl-2, and BAX, inducing binary modes: malignant and benign progression. $W_S = [S^m, S^M]$ = a window of bi-stability. (D) Schematic of anti-apoptosis (\mathbb{P}_t) and apoptosis (\mathbb{P}_a) regions in the $B - X$ plane. Parameters: $J = 0$. Other parameters are given in Table I.

4(A-C). [page 9, Fig 3; page 10, Fig 4, in the (marked) revised manuscript]

- How authors derived optimal control objective function equation (15)? What kind of numerical solution technique was used to solve the equation?

(Response): Thanks for your valuable comment. We applied optimal control theory to find the “best” treatment regime that minimizes the tumor volume under the certain level of tumor concentration and the amount of the drugs used. The objective function in Eq. (15) consists of three parts: reducing the tumor volume in a certain level, remaining in an apoptosis state ($\{(B, X) : B < th_B, X > th_X\}$), and minimizing the use of drugs (u_S and u_D). We used quadratic forms to simplify analysis with the convexity properties which are common in control problems of biological models [45]. For the controls in the integrand, we added linear terms to regularize the amount of drug used. In general, linear controls are more meaningful

biologically than quadratic, but it is more difficult to analyze the system mathematically. We added the following phrase in the main text:

“We used quadratic forms to simplify analysis with the convexity properties which are common in control problems in biological models [45]. For the controls in the integrand, we added linear terms to regularize the amount of drug used. In general, linear controls are more meaningful biologically than quadratic forms, but it is more difficult to analyze the system mathematically. Weight for each control is provided by ...”

[page 8, 1st paragraph, lines 8-12, in the (marked) revised manuscript]

To obtain the numerical solutions of the control problems, we used Forward-Backward Sweep Method (FBSM) which is based on shooting methods to solve boundary value problems [34]. We added the following statement at the end of the paragraph: “To obtain the numerical solutions of the control problems, we used forward-backward sweep method which is based on shooting methods to solve boundary value problems [34].”

[page 8, 1st paragraph, last two lines, in the (marked) revised manuscript]

4. Between strategy 1 and 2, which one is better in what condition?

(Response): We are not comparing strategy 1 with strategy 2. While we in strategy 1 scheme, we were comparing optimal control strategy with other typical injection strategies (constant and periodic injection), we were comparing various injection combination with optimal control in a very specific case with 3 injections of IFN- β and 3 injections of DDP. In the latter case, we identified the best strategy out of many possible combination. It is hard to compare which one is better between strategy 1 and strategy 2 since it is in a different category.

5. Discussion of implication, merit, a couple of limitation, future outlook would be helpful.

(Response): Thank you for suggestions. For discussion of implication and merit, we added the following sentence in Conclusion Section

“The mathematical model in this study may provide a comprehensive understanding of the IFN- β /JAK-induced STAT signaling network and the associated optimal control method may suggest an optimal infusion strategy of anti-cancer drugs in clinical setting.” [page 19, 1st paragraph, lines 3-6, in the (marked) revised manuscript]

For discussion of a couple of limitation, future outlook, we rewrote and added a paragraph in Conclusion Section as follows:

“Our study has three main limitations:

(i) Delivery of anti-cancer drugs is a complex process including transport

of the anti-cancer agent through tissue. In this work, we did not take into account spatial movement of these agents. Therefore, a more general framework such as partial differential equations (PDEs) instead of the ODE model in this work may better describe the spatial transport of drugs. However, development of an optimal control scheme in the PDE model for a larger multi-scale system including blood vessels is still a challenge and we plan to investigate the spatial aspect of drug transport.

(ii) Our optimal control problems have not considered a linear form of controls which may be more realistic than a quadratic form. In particular, $\int_{t_s}^{t_e} u_S(t)dt$ and $\int_{t_s}^{t_e} u_D(t)dt$ represent the total amount of IFN- β and DDP, respectively. Therefore, this type of control forms in a minimization problem can be clearly interpreted as drug toxicity or cost by introducing weights [34, 45, 46]. We plan to develop a linear form of controls in a feasible setting of tumor models in future work.

(iii) Tumor microenvironment and signaling networks play a major role in cancer progression and invasion. We did not take into account various microenvironmental factors including immune cells (macrophages, neutrophils, T cells, Th cells, T regs, and NK cells), cytokines/chemokines, and inter- and intra-cellular molecules. We plan to study these critical factors in future work.

Our mathematical formulation in this work can contribute to development of a new theoretical approach for other cell killing mechanism such as necroptosis [8, 15, 47] and autophagy [40] in cancer by the optimal control of key regulators in a ODE/PDE model [48, 24, 32, 2, 11] or multi-scale hybrid model [22, 21, 26, 27, 25, 28, 31, 30] where key cellular death programs within the cancer cells can be taken into account at individual cell level in the multi-scale system.”

[page 19, 3rd paragraph - 6th paragraph, in the (marked) revised manuscript]

Minor issues:

1. The model is developed in equation (1) to (4) and then non-dimensionalized in (7) to (10). Further, the model is extended incorporating equations (11) to (14) without discussing the dimensional relevance of the variables and parameters appeared in (11) to (14) with the parameters in equation (6).

(Response): Thank you for pointing out this. We now added the dimensional version of equations in Supplementary Information File and the corresponding nondimensionalization was performed there. We now added the following in the main text:

“A dimensional version of equations of tumor volume, IFN- β , JAK2 and

DDP corresponding to Eqs (13)-(16) was introduced and nondimensionalization was performed in Supplementary Information File.” [page 6, 8th paragraph, lines 1-3, in the (marked) revised manuscript]

We also added the following paragraph in order to provide the connections: “See Table I for parameter values in Eqs. (9)-(16) in a dimensionless form. Parameter values in the dimensional form are listed in Table S1 in the Supplementary Information File. Since our mathematical model contains many known and unknown parameter values, we provided parameter estimation in the Supplementary Information File, which is a necessary step toward fundamental and deep understanding of the dynamical process of the mathematical model. Parameter values are calculated based on empirical data such as half-life or estimated by fitting to experimental observation based on the mathematical structure of our model.” [page 6, 5th paragraph, lines 4 - page 7, 1st paragraph, in the (marked) revised manuscript]

2. Equation (14): $u_D \leftarrow u_D(t)$.

(Response): Thank you for careful reading. Following the reviewer’s suggestion, we now replaced ‘ u_D ’ with ‘ $u_D(t)$ ’ [page 6, Eq (16), in the (marked) revised manuscript]

3. Page 5, line 57 to Page 6, line 53: Weight for each control is provided by parameters C1,C2,C3,C4. Linear ($u_S(t)$) and quadratic ($u_S(t)^2$) forms in Eq. (15) represent the costs. Here, the weights C1,C2,C3,C4 are associated with costs, and $u_S(t)$, $u_D(t)$ should be associated with the doses of injection. Further, as C1,C3 are weight parameters for different degree of the same control, should they be necessarily different?

(Response): In general, if controls have different scales, those weight constants could be different in order to keep the balance of effects on the objective function. For the same reason, C1 (C3) was chosen four times larger than C2 (C4) taking into account the amount of IFN-beta an DDP used. Meanwhile, the weight parameters C1 and C3 in our optimal problems are not necessarily different because, as we mentioned above, the linear terms are used for regularization of the controls.

4. T^* is mentioned in the table but not appeared in the text.

(Response): Thank you for careful reading. T^* indicates the reference value of tumor volume and it did not appear in the main text since we did not provide the nondimensionalization process for tumor volume, IFN- β , JAK2, and DDP. We now added the nondimensionalization process for tumor volume, IFN- β , JAK2, and DDP in Supplementary Information File (SI) and

***T** appears in SI.**

[Supplementary Information File, page 2, ‘Nondimensionalization’ Section, Eq (5) in the (marked) revised manuscript]

5. Please provide the figure captions in detail. For example, the meaning of the light red and light blue shaded region in Fig 3(A,B,C) is not mentioned in the caption. Further, in Fig 4D, the shaded regions are dark. I wonder if the all the shaded regions are same or light or dark shades have different meaning.

(Response): Thank you for suggestions. We now added the following sentence in Fig 3 “Blue region = up-regulation of STAT1 + down-regulation of STAT3, pink region = down-regulation of STAT1 + up-regulation of STAT3.” [page 9, Fig 3 caption, lines 3-4, in the (marked) revised manuscript]

Yes, all the shaded regions are some indeed. We did not notice this subtle difference. Thank you for pointing out this difference. Following the reviewer’s suggestion, we now replaced the dark blue and dark red area with light blue and light pink area in Fig 4 in order to make them consistent. [page 10, Fig 4, in the (marked) revised manuscript]

6. Page 10, line 21: Please check decreased activities of STAT1(A? increased?)/BAX (D? increased) and increased (decreased) levels of STAT3/Bcl-2.

(Response): Thank you for careful reading and unclear expression in the caption of Fig 5. We meant that the down-regulation of STAT1/BAX and up-regulation of STAT3/BAX. Mathematically, it is an increasing function of S in the STAT1 graph for instance but we meant the STAT1 level stayed below the threshold. Same principle was applied to others. We now replaced “decreased activities of STAT1/BAX and increased levels of STAT3/Bcl-2.” with “down-regulation of STAT1/BAX (i.e., $S_1 < S_1^{th}$, $X < X^{th}$, $\forall S$ ($0 \leq S \leq 1$)) and up-regulation of STAT3/Bcl-2 (i.e., $S_3 > S_3^{th}$, $B > B^{th}$, $\forall S$ ($0 \leq S \leq 1$)).” [page 11, caption of Fig 5, lines 2-3, in the (marked) revised manuscript]

7. Page 10, line 33: Here, NS , ND are the total infusion of IFN- β ← Here, NS , ND are the total number of infusion of IFN- β .

(Response): Thank you for careful reading. We now replaced “Here, NS , ND are the total infusion of IFN- β ” with “Here, NS , ND are the total number of infusion of IFN- β .” [page 11, 2nd paragraph, lines 3-4, in the (marked) revised manuscript]

8. Page 11, line 50: tumor size at $t = 30$ days is shown, t is non-dimensional time, but here it is mentioned in a unit of day.

(Response): We appreciate the careful reading. Yes, the time is non-dimensional time. We now removed 'days' throughout the manuscript (Figure caption and main text).

“5.2344 every four days and 24.9251 every two days” with “5.2344 and 53.4109” [page 8, 1st paragraph, line 18, in the (marked) revised manuscript],

“4.1875 every days and 24.9251 every days” with “4.1875 and 24.9251, respectively” [page 8, 1st paragraph, line 19, in the (marked) revised manuscript],

“on 1,3,5,7,9 days” with “at t=1,3,5,7,9” [page 11, last paragraph, line 2, in the (marked) revised manuscript],

“20 days” with “t=20” [page 12, 1st paragraph, lines 1,2, in the (marked) revised manuscript],

“25 days” with “t=25” [page 12, 1st paragraph, lines 8,11, in the (marked) revised manuscript],

“1,3,5,7,9 days” with “t=1,3,5,7,9” [page 12, Fig 6 caption, line 3, in the (marked) revised manuscript],

“[0,30 days]” with “the time interval [0,30]” [page 12, 2nd paragraph, line 1, in the (marked) revised manuscript],

“t=30 days” with “t=30” [page 12, 2nd paragraph, line 4, in the (marked) revised manuscript],

“at the last few days” with “at the last” [page 13, 1st paragraph, line 3, in the (marked) revised manuscript],

removed ‘days’ [page 13, Fig 7(B,C,D) X-axis label; page 14, 1st paragraph, lines 3, 7, in the (marked) revised manuscript],

removed ‘(five days)’ [page 14, 1st paragraph, line 3, in the (marked) revised manuscript],

“4.5 days” with “4.5” [page 14, 1st paragraph, line 7, in the (marked) revised manuscript],

“entire schedule of 30 days (T=30)” with “entire schedule (upto t=30)” [page 15, 2nd paragraph, line 3, in the (marked) revised manuscript],

“IFN- β is administered every 4 days with $u_S = 5.2344$ while DDP is administered every 2 days with $u_D = 53.4109$ ” with “IFN- β is administered with $u_S = 5.2344$ and $\tau_s = 4$ while DDP is administered with $u_D = 53.4109$ and $\tau_d = 2$ ” [page 15, 2nd paragraph, lines 2-3, in the (marked) revised manuscript],

removed ‘days’ [page 16, Fig 10(A,B,C,D,E,G,H) X-axis label, in the (marked) revised manuscript],

9. Page 11, 12, 13: Several ' are converted to ‘.

(Response): Thank you for careful reading. We now replaced ' with ‘. [page 12, 2nd paragraph, line 2 ; page 13, 1st paragraph, lines 1,4,7, and

15; page 14, 1st paragraph, lines 2,3,7, and 8; page 14, 2nd paragraph, line 6; page 17, 1st paragraph, lines 4-19; page 18, Fig 12 caption, lines 2,4, in the (marked) revised manuscript]

10. Page 14, line 45: following two cases are considered: ← following two strategies are considered:

(Response): Thank you for nice suggestion. Following the reviewer's suggestion, we now replaced

“following two cases are considered:” with “following two strategies are considered:” [page 15, 2nd paragraph, line 3, in the (marked) revised manuscript]

11. Page 14, line 45: Strategy I: control for the entire schedule of 30 days ($T = 30$). Here, time is denoted by T , which is the symbol for tumor volume.

(Response): Thank you for careful reading. We replaced

“entire schedule of 30 days ($T = 30$).” with “entire schedule (upto $t = 30$).” [page 15, 2nd paragraph, line 3, in the (marked) revised manuscript]

12. Page 14, line 54,55: (1, 0, 0, 4d-5, 1d-5, 4d-5, 1d-5, 0). Does the 1d-5 means 1e-5 here?

(Response): Thank you for careful reading. Yes, indeed, it is the case. We replaced “(1, 0, 0, 4d-5, 1d-5, 4d-5, 1d-5, 0)” with “(1, 0, 0, 4e-5, 1e-5, 4e-5, 1e-5, 0)” [page 15, last paragraph, lines 6-7, in the (marked) revised manuscript]

We also replaced “(12, 1, 4, 2d-2, 0, 2d-2, 0, 1.7634)” with “(12, 1, 4, 2e-2, 0, 2e-2, 0, 1.7634)” [page 17, 1st paragraph, lines 3, in the (marked) revised manuscript]

13. **In addition, we replaced “total dose T_D26 over one injection period (five days)” with “the relatively smaller total dose ($T_D = 26$)”** [page 14, 1st paragraph, lines 2-3, in the (marked) revised manuscript],
“the total dose T_D41 ” with “the higher total dose ($T_D = 41$)” [page 14, 1st paragraph, line 4, in the (marked) revised manuscript]

REFEREE 2:

This paper addresses intracellular signal transduction systems, including the STAT family. In particular, the control of the important bistable structures of cancer's apoptotic and anti-apoptotic states is investigated. This paper is the first to study the mechanism of apoptosis via the JAK-STAT signaling pathway using optimal control theory. As a therapeutic application, this study aims to find the optimal injection scheme of combination therapy (IFN- β + DDP) by optimal control theory. The model predicted that an optimally controlled schedule of these two drugs could provide better antitumor effects at minimal cost.

Both the methods and results are new and valuable, and I recommend accepting this article for publication.

1. [Page 4; Line 13; Equation (5)] Should $f_2(s)$ be $f_2(s, j)$?

(Response): Thank you for careful reading. Indeed, it was a typo and $f_2(s)$ should be $f_2(s, j)$. We now replaced $f_2(s)$ with $f_2(s, j)$ [page 5, 1st paragraph, Eq (7), in the (marked) revised manuscript]

REFEREE 3:

In general the background referencing is incomplete. Researchers other than Kim et al. have done considerable work in modeling the Bcl-2 mediated regulation of apoptosis in cancer cells.

(Response): Thank you for careful reading and suggestions. We now added the following phrase in Introduction Section

For example, mathematical models of Bcl-2 signaling networks illustrated importance of molecular play including intrinsic Bcl-2 apoptosis pathway [35, 5, 36, 10, 13], bistability in apoptosis [3], interaction between p53 and Bcl-2 [9], VEGF-Bcl-2 in angiogenesis [17, 18], and MOMP regulation in pattern recognition [50]. See reviews in [52, 19, 16, 42] for systems-based approaches of Bcl-2 and cell-death program. [page 2, last paragraph, line 4 - page3, 1st paragraph, lines 1-4, in the (marked) revised manuscript]

The Introduction is a bit confusing. It will help eg at the end of the first paragraph to highlight knowledge gaps that mathematical modeling can fill. What is the need or motivation for the model?

(Response): Thank you for suggestions. We restructured the Introduction Section and added the following phrases:

Despite previous studies of apoptotic signaling, fundamental mechanism of the JAK-IFN β -mediated apoptosis processes is poorly understood. However, translational studies with experimental data [38] support the considerable benefits of apoptosis-based therapy [12]. Qualitative analysis may contribute to fundamental understanding of this complex system. In particular, a new approach may identify the key functions and regulation of both JAK and IFN β -mediated STATs in the apoptosis pathways within cancer cells.

[page 2, 1st paragraph, lines 20–25, in the (marked) revised manuscript]

At the end of the introduction it is not clear what the aim is. What will be achieved? What is new, in terms of the model? I get no sense of what the paper really accomplishes.

(Response): Thank you for suggestions. Nobody studied JAK2-STATs-Bcl-2-BAX signaling network in regulation of cancer killing, the mathematical model itself and optimal control application, up to our knowledge. We now replaced (i) the regulatory control of apoptotic pathways through IFN- β and JAK2. (ii)

how changes in Bcl-2 and BAX affect cancer progression, ” with “(i) unexplored structure of the STAT-JAK2-Bcl-2-BAX signaling pathways, (ii) how changes in IFN- β , STATs, and JAK2 affect cancer progression,” [page 3, 2nd paragraph, lines 4–5, in the (marked) revised manuscript]

We also added the following phrases at the end of the Introduction Section:
 “We found that JAK2 and mutual antagonism between STAT1 and STAT3 play a major role in regulation of the apoptosis and anti-apoptotic status in cancer cells, thus tumor growth dynamics, and obtained the optimal injection strategies of both JAK2 inhibitors and IFN- β by minimizing costs and maximizing anti-tumor efficacy through an optimal control theory. ” [page 3, 2nd paragraph, lines 6–9, in the (marked) revised manuscript]

So many parameters have been Est (see Table 1) that one cannot really have confidence in them. Estimated from where? Are these fit to some data? Why are most of them 1?

(Response): Thank you for careful reading. Some of parameters such as decay rates are calculated from biological observation such as half-life of molecules. Many of estimated parameters came from fitting the simulation results to experimental data, for instance responses (up- and down-regulation) of intracellular molecules in response to periodic injection of IFN- β . Some of parameters are 1 because the mathematical structure of the mathematical model in a dimensionless form can gives us nice mathematical advantages to represent the biological behaviors such as switching of states in a given text. We now added the following in Supplementary Information File as well:

“ λ_1, λ_2 : By using the reference value of Bcl-2 ($B^* = 10nM$), and taking the signal source of the Bcl-2, $f_3 = 5.78 \times 10^{-2}nM/h$, we estimate the dimensionless value of signaling strength to the Bcl-2 complex to be $\lambda_1 = \frac{f_3}{\mu_{S_1}B^*} = 0.2$. Similarly, by taking the reference value of BAX ($X^* = 351\mu M$), and taking the signal source of BAX, $f_4 = 2.0288\mu M/h$, we estimate the dimensionless value of signaling source to be $\lambda_2 = \frac{f_4}{\mu_{S_1}X^*} = 0.2$.

λ_3, k_5 : We assume the autocatalytic rate (a_7) of Bcl-2 is same as its negative contribution ($\mu_{S_1}B^* = (2.89 \times 10^{-2}h^{-1}) \times (1.0 \times 10^{-2}\mu M) = 2.89 \times 10^{-4}\mu M/h$), leading to the dimensionless parameter value $k_5 = \frac{a_7}{\mu_{S_1}B^*} = 1.0$. By assuming that the signaling rate ($\lambda_{STAT3} \times (1.38\mu g/mL)$) from STAT3 is at least 1.2-fold larger than its negative contribution ($\mu_{S_1}B^*$) and by taking $\lambda_{STAT3} = 2.513 \times 10^{-4}\mu M \cdot mL/(h \cdot \mu g)$, we get the dimensionless parameter value $\lambda_3 = \frac{\lambda_{STAT3}S_3^*}{\mu_{S_1}B^*} = \frac{\lambda_{STAT3} \times (1.38\mu g/mL)}{(2.89 \times 10^{-2}h^{-1}) \times (1.0 \times 10^{-2}\mu M)} = 1.2$.

k_1, k_3, k_7 : We take the dimensionless parameter value $k_1 = \frac{a_1}{\mu_{S_1}S_1^*} = 4.0$ by assuming that the autocatalytic rate (a_1) of STAT1 is at least 4-fold larger than its negative contribution ($\mu_{S_1}S_1^*$) from its decay in the absence of the inhibitory pathway from the STAT3. In a similar fashion, we get the autocatalytic rate of STAT3 (k_3) and BAX (k_7), $k_3 = \frac{a_4}{\mu_{S_1}S_3^*} = 4.0$ and $k_7 = \frac{a_{10}}{\mu_{S_1}X^*} = 4.0$.

k_2, k_4, k_6, k_8 : The Hill-type coefficients a_2, a_5, a_8, a_{11} (k_2, k_4, k_6, k_8 : parameters with

dimensionless form) are fixed to be unity.

k_9, k_{10}, μ_T : In the absence of apoptosis, the carrying capacity of cancer is T_0 . However, in the presence of apoptosis, the carrying capacity is $T_0 \left(1 - \frac{\mu_T}{r \left(1 - \frac{k_9 S_1^2}{k_{10}^2 + S_1^2} \right)} \right)$.

Our system assumes that most cancer cells die by the apoptosis program. We set $k_9 = 1.0, k_{10} = 10$, and $\mu_T = 0.1$ and the high level of STAT1 is about 4.5. Then, the steady state value of tumor volume in the presence of apoptosis is smaller than equilibrium value in the absence of apoptosis, $T_0 \left(1 - \frac{\mu_T}{r \left(1 - \frac{k_9 S_1^2}{k_{10}^2 + S_1^2} \right)} \right) \approx 0.01 \times T_0$.

The list of dimensionless parameters and their values are given in Table I in the main text. ”

[Supplementary Information File, 7th paragraph in page 1 - 4th paragraph in page 2, in the (marked) revised manuscript]

1. [Abstract] Doesn't really mention what is novel about this work. Also should be mathematically more precise, eg, what is being optimized, what is the constraint?

(Response): Thank you for nice suggestions. We now replaced

“The mathematical model unveils the structure and functions of the intracellular signaling and cellular outcomes of the anti-tumor drugs in the presence of IFN- β and JAK stimuli. We finally use an optimal control theory in order to suggest optimal anti-tumor efficacy with minimal costs in response to drug administration.” **with** “The roles of JAK-STATs signaling in regulation of the cell death program in cancer cells and tumor growth are poorly understood. The mathematical model unveils the structure and functions of the intracellular signaling and cellular outcomes of the anti-tumor drugs in the presence of IFN- β and JAK stimuli. We identify the best injection order of IFN- β and DDP among many possible combinations, which may suggest better infusion strategies of multiple anti-cancer agents at clinics. We finally use an optimal control theory in order to maximize anti-tumor efficacy and minimize administrative costs. In particular, we minimize tumor volume and maximize the apoptotic potential by minimizing the Bcl-2 concentration and maximizing the BAX level while minimizing total injection amount of both IFN- β and JAK2 inhibitors (DDP).” [Abstract, lines 7–17, in the (marked) revised manuscript]

2. [Abstract] Line 11 what specifically is meant by immunity?

(Response) In order to make it clear, we replaced “immunity” with “various challenges faced by the immune system” [Abstract, lines 1–2, in the (marked) revised manuscript]

3. [Abstract] Line 13 anti-apoptosis isn't really cellular fate.

(Response): Thank you for careful reading. We now replaced “cellular fate” with “cellular states” [Abstract, lines 3–4, in the (marked) revised manuscript]

4. [Introduction] Paragraph 1 two different dichotomous states of cell death: (i) an apoptosis progression and (ii) an anti-apoptosis progression. This is confusing. What is meant by anti-apoptosis progression? Does this also lead to cell death? If not, then how can it be a state of cell death?

(Response): Thank you for pointing out this. Following the reviewer's suggestion, we now replaced “dichotomous states of cell death: (i) an apoptosis progression and (ii) an anti-apoptosis progression.” with “dichotomous states: (i) an apoptosis progression and (ii) an anti-apoptosis state (inactivation of cell death program).” [page 2, 1st paragraph, lines 9–10, in the (marked) revised manuscript]

5. [Introduction] Paragraph 1 What is meant by apoptosis-mediated therapy? This seems to be poorly worded.

(Response): Thank you for pointing out this. More commonly used term is 'apoptosis-based therapy'. We now replaced “apoptosis-mediated therapy” with “apoptosis-based therapy [12]” [page 2, 1st paragraph, line 22, in the (marked) revised manuscript] in the edited phrases in 1st paragraph as following: “Despite previous studies of apoptotic signaling, fundamental mechanism of the JAK-IFN β -mediated apoptosis processes is poorly understood. However, translational studies with experimental data [38] support the considerable benefits of apoptosis-based therapy [12]. Qualitative analysis may contribute to fundamental understanding of this complex system. In particular, a new approach may identify the key functions and regulation of both JAK and IFN β -mediated STATs in the apoptosis pathways within cancer cells.”

[page 2, 1st paragraph, lines 20–25, in the (marked) revised manuscript]

6. [Introduction] Paragraph 2 ‘However, there was no study of apoptosis mechanism through JAK-STAT signaling pathway with optimal control theory.’ This sentence has a couple of issues. First, it is poorly worded. Second, it is oddly specific. Is the claim that no mathematical studies of JAK-STAT mediation of apoptosis have been published? Or that no mathematical studies of JAK-STAT mediation of apoptosis using in some manner optimal control have been published? If the latter is the case, then this is hardly a novel study. In my experience, most mathematical models of cancer response to therapy include some manner of treatment optimization, whether or not they use optimal control.

(Response): Thank you for pointing out this. It's indeed the latter case. We replaced

“In particular, optimal control approaches are used to identify optimal schedule of anti-cancer drugs [2, 20, 43, 44]. However, there was no study of apoptosis mechanism through JAK-STAT signaling pathway with optimal control theory.” with “In particular, optimal control approaches are used to identify optimal schedule of anti-cancer drugs targeting stromal/immune cells and various signaling pathways [2, 20, 43, 44]. Fundamental mechanism of the JAK-STAT-mediated cancer cell killing is still poorly understood. To our knowledge, no mathematical study has investigated the underlying mechanisms of JAK-STAT mediation of apoptosis in cancer cells.” [page 3, 1st paragraph, lines 4–8, in the (marked) revised manuscript]

7. [Introduction] Fig 1 should be referenced in paragraph 1 where the biology is introduced.

(Response): Thank you for careful reading. We now added the following in paragraph 1:

“See Fig 1.” [page 2, 1st paragraph, line 11, in the (marked) revised manuscript]

8. [Methodology] Paragraph 1 why is time t with an overbar? This is not standard and what is the motivation?

(Response): In mathematics, we typically use variable symbols with an overbar in order to indicate they are variables with dimension (sometimes dimensionless variables depending on authors and styles). Sometimes, some researchers use a ‘tilde’ instead of an overbar. As we explained below, we are using many symbols for dimensional or dimensionless variables and parameters. Therefore, this was our choice to make it more clear rather than making it confused by introducing so many different symbols. If we introduce too many symbols without using bar, we would be running out of symbols and making it confusing.

9. [Methodology] Where do the equations (1) - (4) come from? That is, why these functional form choices? Have they been derived from first principles? Are they purely phenomenological? How are they justified?

(Response): The equations (1) - (4) are based on mass balance equation. Since, we don't use partial differential equation, the simple mass balance concept 'input - output = accumulation' concept is used, i.e., $\frac{dy}{dt} = \text{In} - \text{Out}$. Let's suppose that we have a system of ODEs with N variables ($y_i = y_i(t)$, ($i = 1, \dots, N$)). In general, the mass balance of given intracellular

variable $y_i = y_i(t)$, ($i = 1, \dots, N$) is used to derive governing equation

$$\frac{dy_i}{dt} = f_i(\mathbf{y}) + g_i(\mathbf{y}) - h_i(\mathbf{y}), \quad (15)$$

where $\mathbf{y} = (y_1, y_2, \dots, y_N)$, the function $f_i(\mathbf{y})$ represents the source, $g_i(\mathbf{y})$ represents inhibition, and $h_i(\mathbf{y})$ represents outflux due to natural decay, i.e. $h_i(\mathbf{y}) = \mu_i y_i$, where μ_i is the decay rate. The source function $f_i(\mathbf{y})$ can be described below based on biological observations. A fractional form for the inhibition term in Eq (17) was chosen as the qualitative representation of negative feedbacks in this work. Specifically, we use the form

$$g_i(\mathbf{y}) = \frac{\zeta_1 \zeta_2^n}{\zeta_2^n + \alpha_i F(y_j)} \quad (16)$$

for autocatalytic activity with the inhibition process of the intracellular variable y_i by another intracellular variable y_j ($i \neq j$), where ζ_1, ζ_2 are constants, the parameter α_i represent the inhibition strength along with amount of the variable y_j via a function $F(y_i)$ ($\mu_i, \zeta_1, \zeta_2, \alpha_i \in \mathbb{R}^+, n \in \mathbb{Z}^+$). In the absence of source, this inhibition term with the decay term, $-\mu_i y_i$ provides the baseline concentration $y_i^* \approx \frac{\zeta_1}{\mu_i}$ of the given molecule y_i at a steady state when the inhibition strength $F(y_j)$ is small or zero. (When $f_i(\mathbf{y}) \neq 0$, the baseline becomes $y_i^* \approx \frac{f_i + \zeta_1}{\mu_i}$.) The relative balance between the source term and inhibition strength from y_j essentially determines the concentration of the molecule y_i . Thus, by comparing the simulated y_i level with experimental data in the presence and absence of the inhibitory molecule y_j in the system, one can build a mathematical model in Eq (17) with the consistent, up- or down-regulated y_i . Several studies [1, 29, 23, 22] have shown that this fractional form for the negative feedbacks may reproduce analytic structure of genetic networks (positive and negative feedbacks) and qualitative dynamics such as bi-stability with experimental validation. Other forms of negative feedbacks (eg. one based on chemical reactions) have been used in the literature [49, 25]. Essentially, this choice allows us to model the inhibition process between intracellular molecules, which was observed in experiments. Other forms are possible, of course. However, other choices typically involves countless chemical reactions and another choices of model design based on chemical reactions at some point when one wants to model the negative feedback of the given molecule by another, which generates same degrees of uncertainty, in other words every model design have their own assumptions. In order to make it clear, we now replaced

“The mass balance..” with

“In this work, we ignore any spatial effects on dynamics of given system. In general, the mass balance of given intracellular variable $y_i = y_i(t)$, ($i = 1, \dots, N$) is used to derive governing equation

$$\frac{dy_i}{dt} = f_i(\mathbf{y}) + g_i(\mathbf{y}) - h_i(\mathbf{y}), \quad (17)$$

where $\mathbf{y} = (y_1, y_2, \dots, y_N)$, the function $f_i(y)$ represents the source, $g_i(y)$ represents inhibition, and $h_i(y)$ represents outflux due to natural decay, i.e. $h_i(y) = \mu_i y_i$, where μ_i is the decay rate. The source function $f_i(y)$ can be described below based on biological observations. A fractional form for the inhibition term in Eq (17) was chosen as the qualitative representation of negative feedbacks in this work. Specifically, we use the form

$$g_i(\mathbf{y}) = \frac{\zeta_1 \zeta_2^n}{\zeta_2^n + \alpha_i F(y_j)} \quad (18)$$

for autocatalytic activity with the inhibition process of the intracellular variable y_i by another intracellular variable y_j ($i \neq j$), where ζ_1, ζ_2 are constants, the parameter α_i represent the inhibition strength along with amount of the variable y_j via a function $F(y_j)$ ($\mu_i, \zeta_1, \zeta_2, \alpha_i \in \mathbb{R}^+, n \in \mathbb{Z}^+$). In the absence of source, this inhibition term with the decay term, $-\mu_i y_i$ provides the baseline concentration $y_i^* \approx \frac{\zeta_1}{\mu_i}$ of the given molecule y_i at a steady state when the inhibition strength $F(y_j)$ is small or zero. (When $f_i(\mathbf{y}) \neq 0$, the baseline becomes $y_i^* \approx \frac{f_i + \zeta_1}{\mu_i}$.) The relative balance between the source term and inhibition strength from y_j essentially determines the concentration of the molecule y_i . Thus, by comparing the simulated y_i level with experimental data in the presence and absence of the inhibitory molecule y_j in the system, one can build a mathematical model in Eq (17) with the consistent, up- or down-regulated y_i . Several studies [1, 29, 23, 22] have shown that this fractional form for the negative feedbacks may reproduce analytic structure of genetic networks (positive and negative feedbacks) and qualitative dynamics such as bi-stability with experimental validation. Other forms of negative feedbacks (eg. one based on chemical reactions) have been used in the literature [49, 25]. Then, the mass balance..”

[From page 3, line (-2) to page 4, 1st paragraph, line 21, in the (marked) revised manuscript]

10. [Methodology] In general, it would be helpful to explain what the terms on the right hand side are doing rather than get into listing every parameter. Also, please include in the main text, a table with model variables and parameters and all units.

(Response): Thank you for suggestions. We now added the explanatory words under each term in Eqs (3)-(6), (9)-(12), (13)-(16) in order to explain the terms on the right hand side of each equation. In addition, we replaced

“where $f_1(s)$ is a signaling function of IFN- β (s) to the STAT1, and $f_2(s, j)$ is a signaling function of IFN- β (s) and JAK2 (j) to the STAT3. The parameters f_3 and f_4 represent the signaling source of Bcl-2 and BAX, respectively. The autocatalytic activity parameters for the modules of STAT1, STAT3, Bcl-2, and BAX are denoted by a_1, a_4, a_7 , and a_{10} , respectively. The parameters a_2, a_5, a_8 , and a_{11} are the Hill-type inhibition saturation constants from the counter part of STAT1, STAT3, Bcl-2, and BAX, respectively. There are

four inhibition strengths: a_3 for STAT1 by STAT3, a_6 for STAT3 by STAT1, a_9 for Bcl-2 by STAT1, and a_{12} for BAX by Bcl-2. The parameter λ_{STAT3} is the activation rate of Bcl-2 by STAT3. The parameters $\mu_{S_1}, \mu_{S_3}, \mu_{Bcl2}$, and μ_{BAX} are the clearance/death rate of STAT1, STAT3, Bcl-2, and BAX, respectively. We set $\mu_{S_1} = \mu_{S_2}$ due to the same half-life of STAT1 and STAT3 (Supplementary Information File).”

with

“where s, j are concentrations of IFN- β and JAK2, respectively. The rate of changes in STAT1 involves the signal source from IFN- β via a function $f_1(s)$, autocatalytic activity with inhibition from STAT3 ($\bar{S}_3 \dashv \bar{S}_1$) and natural decay at a rate μ_{S_1} . In particular, the general form in Eq. (2) is used for the autocatalytic activity/inhibition in the second term on the right hand side (RHS) of Eq. (3) with the autocatalytic activity parameter a_1 , the Hill-type inhibition saturation constants a_2 , and inhibition strength a_3 . Consistent forms and parameter notations were used for autocatalytic activity with inhibition of STAT3 (a_4, a_5, a_6), Bcl-2 (a_7, a_8, a_9), and BAX (a_{10}, a_{11}, a_{12}) in the second terms in Eqs. (4)-(6). In a similar fashion, STAT3 in Eq. (4) undergoes the signaling from both JAK and IFN- β via a function $f_2(s, j)$, autocatalytic activity with inhibition from STAT1 ($\bar{S}_1 \dashv \bar{S}_3$) and natural decay at a rate μ_{S_3} . On the other hand, Bcl-2 in Eq. (5) is regulated by the signal source at a fixed rate f_3 , autocatalytic activity with inhibition from STAT1 ($\bar{S}_1 \dashv \bar{B}$), up-regulation from STAT3 ($\bar{S}_3 \rightarrow \bar{B}$) at a rate λ_{STAT3} , and natural decay at a rate μ_{Bcl2} . Finally, BAX in Eq. (6) is regulated by the signal source at a fixed rate f_4 , autocatalytic activity with inhibition from STAT1 ($\bar{B} \dashv \bar{X}$), and natural decay at a rate μ_{BAX} . We set $\mu_{S_1} = \mu_{S_2}$ due to the same half-life of STAT1 and STAT3 (Supplementary Information File).”

[From page 4, last paragraph below Eq (6), in the (marked) revised manuscript]

We also added the following for the equation of the tumor cell volume

“Here, the first term on RHS of Eq (13) represents the STAT1-controlled growth of tumor cells. On the other hand, the second term represents the programmed cell death of tumor cells when the intracellular signaling induces the apoptosis in response to external stimuli such IFN- β .”

[page 6, 2nd paragraph, lines 8–12, in the (marked) revised manuscript]

For the equation of the IFN- β concentration, we replaced “where μ_S is the decay rate of IFN- β .” **with** “The first and second terms on RHS of Eq (14) represent the injection of IFN- β via a function $u_S(t)$ and decay process at a rate μ_S , respectively.” [page 6, 3rd paragraph, lines 8–9, in the (marked) revised manuscript]

For the equation of the concentrations of JAK2 and DDP, we replaced “where J_s is the signaling source of JAK2, μ_J, μ_D are the decay rates of JAK2 and DDP, respectively, and γ_D is the DDP-mediated degradation

constant of JAK. u_D will be set similarly to u_S .” with “JAK2 in Eq (15) undergoes production at a rate J_s , the DDP-mediated degradation of JAK by DDP (γ_D), and decay process at a rate μ_J in the first, second, and third terms on RHS, respectively. The first and second terms on RHS in Eq (16) represent the time-dependent injection of DDP via a function $u_D(t)$ and decay process at a rate μ_D , respectively.” [page 6, 4th paragraph, lines 7–10, in the (marked) revised manuscript]

11. [Methodology] What is the difference between capital S (see paragraph 3 of introduction) and lowercase s (Page 5, lines 8-9)? What do you mean by IFN activity? This does not have any obvious mathematical meaning. Do you mean concentration of IFN? What units?

(Response): Thank you for careful reading and pointing out vague expression. As it was shown in Mathematical model section (under Methodology section), s indicates the IFN- β concentration in a dimensional form and S represents the same concentration in a dimensionless form. We realize that introduction of S in Introduction Section before it is defined in the next section is not appropriate. We replaced “involving eight intracellular variables: STAT1 (S_1), STAT3 (S_3), Bcl-2 (B), BAX (X), IFN- β (S), JAK2 (J), DDP (D), and tumor (T).” with “involving eight variables: concentrations of STAT1, STAT3, Bcl-2, BAX, IFN- β , JAK2, and DDP and tumor volume.” [page 3, 2nd paragraph, lines 2-3, in the (marked) revised manuscript] and removed all little module boxes (Module ‘ S'_1 & ‘ S'_3 , Module ‘ B ’, and Module ‘ X ’) in the middle panel in Fig 1 in order to avoid any confusion. [page 2, middle panel, Fig 1, in the (marked) revised manuscript]

Yes, ‘IFN activity’ means ‘high concentration of IFN- β ’ with a typical unit (g/mL or μM) (these two words are almost synonym in biology since it is sometimes hard to measure concentration of intracellular (or extracellular) molecules). In order to make it clear, we made the following changes:

- (i) “where” \rightarrow “where s, j are concentrations of IFN- β and JAK2, respectively,” [page 4, line 1 after Eq (6), in the (marked) revised manuscript]
- (ii) “IFN- β activity (s) up-regulates the STAT1 level with $f_1(s)$, while STAT3 activity inhibits the STAT1 level with $F_1(\bar{S}_3)$. These functions are assumed that ” \rightarrow “high concentration of IFN- β (s) up-regulates the STAT1 level through the positive function $f_1(s)$, while the high concentration of STAT3 inhibits the STAT1 level through the positive function $F_1(\bar{S}_3)$. In other words, we have mathematical conditions: ” [page 5, 1st paragraph, lines 1-3, in the (marked) revised manuscript]
- (iii) “the STAT3 level while $f_2(s, j)$ is suppressed by the IFN- β .” \rightarrow “the STAT3 level through JAK as well as suppression of the STAT3 level by the IFN- β .” [page 5, 1st paragraph, lines 4-5, in the (marked) revised manuscript]

12. [Methodology] Page 5, Lines 47-48 I do not know what you mean by “in genetic clas-

sification”. Further in this paragraph, what biological observation? From where? In what cells? Is this data published somewhere?

(Response): Thank you for careful reading and suggestions. We replaced “another one (th_X) in genetic classification, in other words, when the mathematical condition $\{(B, X) : B < th_B, X > th_X\}$ is satisfied. The threshold values were set based on biological observation and dynamical system of Eq. (9)-(12).”

with

“another one (th_X), in other words, when the condition $\{(B, X) : B < th_B, X > th_X\}$ is satisfied, as suggested in experiments and modeling works [14, 4, 33, 3, 51, 39, 24]. The threshold values were set based on biological observation [14, 4, 33, 3, 51, 39] and dynamical system of Eq. (9)-(12).” [page 5, 3rd paragraph, lines 4-7, in the (marked) revised manuscript]

13. [Methodology] In equation (11) does the proliferation become negative when the apoptosis condition is satisfied? Then why have a separate apoptosis term? If not, then why does proliferation rate depend on apoptosis threshold? Should it not have its own threshold? Where are the biological data justifying this? Is equation (11) also non-dimensional?

(Response): Yes, the equation (11) (now Eq (13) in the revised version) is non-dimensional. The nondimensionalization of these variables (tumor volume, IFN- β , DDP, JAK) was provided in Supplementary Information File. In equation (11) (now Eq (13) in the revised version), the proliferation does not become negative when the apoptosis condition is satisfied. Even when the proliferation term is zero in the first term on the right hand side (RHS), for example when $r = 0$, when the apoptosis condition is satisfied (i.e., when $I_{\{B < th_B, X > th_X\}} = 1$), the equation becomes

$$\frac{dT}{dt} = - \underbrace{\mu_T T}_{\text{apoptosis}} .$$

and the solution of the equation is $T(t) = \chi_0 e^{-\mu_T t}$ with the initial tumor size χ_0 ($\chi_0 > 0$), thus the tumor size is always non-negative ($T(t) = \chi_0 e^{-\mu_T t} > 0, \forall t \geq 0$). In the original form when the apoptosis condition is satisfied (i.e., when $I_{\{B < th_B, X > th_X\}} = 1$), the middle part in the first term $\left(1 - \frac{k_9 S_1^2}{k_{10}^2 + S_1^2} I_{\{B < th_B, X > th_X\}}\right) = \left(1 - \frac{k_9 S_1^2}{k_{10}^2 + S_1^2}\right) \geq 0$ since $0 \leq \frac{k_9 S_1^2}{k_{10}^2 + S_1^2} \leq k_9, \forall S_1$ and we assumed $k_9 \leq 1$, in particular we set $k_9 = 1$. Therefore, the tumor size is always positive. In order to make this clear, we added the following statement:

“We note that the inhibition part in the middle of the first term $\left(1 -$

$\frac{k_9 S_1^2}{k_{10}^2 + S_1^2} I_{\{B < th_B, X > th_X\}} \geq 0, \forall S_1$ due to our assumption $k_9 \leq 1$. In particular, we set $k_9 = 1$.” [page 6, 2nd paragraph, lines 8-10, in the (marked) revised manuscript]

Inhibition of tumor growth by STAT1 in TME was observed in literature including [6] as described in the paragraph. We have the separate apoptosis term since tumor cells can be killed under the apoptosis condition even when the STAT1 level is low. Switching of the STAT1-mediated inhibition of tumor growth is modeled by adapting the Hill type function $\frac{k_9 S_1^2}{k_{10}^2 + S_1^2}$ rather than using threshold value of STAT1. This Hill type function has similar effect as threshold values. Therefore, those two mechanisms generate combined tumor cell killing in our modeling framework.

14. [Methodology] Page 6, Lines 23-24 what is meant by alternating injection, ie, alternating with what? By the way, this notation is very confusing. Stats are denoted by S_i and drug is also S ?

(Response): Thank you for careful reading and suggestion. However, we have lots of variables and parameters and the labels of variables are limited. Many of labels are already taken. We think that if we introduce another variable for IFNbeta (now with notation ‘S’) etc, it can cause more confusion.

15. [Methodology] Page 6, Lines 29-31 Not clear at all what is meant by this: Dose of DDP is determined on covering section in the neighborhood of injection spot [68] due to different anti-tumor efficacy in the chemotherapy

(Response): Thank you for careful reading. We agree that the statement is not clearly expressed. We now removed the sentence in question from the main text. So, we replaced “Dose of DDP is determined on covering section in the neighborhood of injection spot [7] due to different anti-tumor efficacy in the chemotherapy [37, 41]. The governing” with “The governing” [page 6, 4th paragraph, lines 3-4, in the (marked) revised manuscript]

(etc) Overall, I suggest making the discussion more structured to get your points across more clearly.

(Response): Thank you for suggestions. We made the discussion more structured to get the points across more clearly as follows:

“The mathematical model in this study may provide a comprehensive understanding of the IFN- β /JAK-induced STAT signaling network and the associated optimal control method may suggest an optimal infusion strategy of anti-cancer drugs in clinical setting.” [page 19, 1st paragraph, lines 3-6, in the (marked) revised manuscript]

“Our study has two main limitations:

(i) Delivery of anti-cancer drugs is a complex process including transport of the anti-cancer agent through tissue. In this work, we did not take into account spatial movement of these agents. Therefore, a more general framework such as partial differential equations (PDEs) instead of the ODE model in this work may better describe the spatial transport of drugs. However, development of an optimal control scheme in the PDE model for a larger multi-scale system including blood vessels is still a challenge and we plan to investigate the spatial aspect of drug transport.

(ii) Our optimal control problems have not considered a linear form of controls which may be more realistic than a quadratic form. In particular, $\int_{t_s}^{t_e} u_S(t)dt$ and $\int_{t_s}^{t_e} u_D(t)dt$ represent the total amount of IFN- β and DDP, respectively. Therefore, this type of control forms in a minimization problem can be clearly interpreted as drug toxicity or cost by introducing weights [34, 45, 46]. We plan to develop a linear form of controls in a feasible setting of tumor models in future work.

(iii) Tumor microenvironment and signaling networks play a major role in cancer progression and invasion. We did not take into account various microenvironmental factors including immune cells (macrophages, neutrophils, T cells, Th cells, T regs, and NK cells), cytokines/chemokines, and inter- and intra-cellular molecules. We plan to study these critical factors in future work.

Our mathematical formulation in this work can contribute to development of a new theoretical approach for other cell killing mechanism such as necroptosis [8, 15, 47] and autophagy [40] in cancer by the optimal control of key regulators in a ODE/PDE model [48, 24, 32, 2, 11] or multi-scale hybrid model [22, 21, 26, 27, 25, 28, 31, 30] where key cellular death programs within the cancer cells can be taken into account at individual cell level in the multi-scale system.”

[page 19, 3rd paragraph - 6th paragraph, in the (marked) revised manuscript]

References

- [1] Aguda, B., Kim, Y., Hunter, M., Friedman, A., and Marsh, C. (2008). MicroRNA regulation of a cancer network: Consequences of the feedback loops involving miR-17-92, E2F, and Myc. *PNAS*, 105(50):19678–19683.
- [2] Aspirin, A., de Los Reyes V, A., and Kim, Y. (2021). Polytherapeutic strategies with oncolytic virus-bortezomib and adjuvant nk cells in cancer treatment. *J R Soc Interface.*, 18(174):20200669.
- [3] Bagci, E., Vodovotz, Y., Billiar, T., Ermentrout, G., and Bahar, I. (2006). Bistability in apoptosis: roles of bax, bcl-2, and mitochondrial permeability transition pores. *Biophys J.*, 90(5):1546–1559.
- [4] Campbell, K. and Tait, S. (2018). Targeting BCL-2 regulated apoptosis in cancer. *Open Biol.*, 8(5):180002.
- [5] Chen, C., Cui, J., Zhang, W., and Shen, P. (2007). Robustness analysis identifies the plausible model of the bcl-2 apoptotic switch. *FEBS Lett.*, 581(26):5143–5150.
- [6] Chen, J., Zhao, J., Chen, L., Dong, N., Ying, Z., Cai, Z., Ji, D., Zhang, Y., Dong, L., Li, Y., Jiang, L., Holtzman, M., and Chen, C. (2015). Stat1 modification improves therapeutic effects of interferons on lung cancer cells. *J Transl Med.*, 13:293.
- [7] de Jongh, F., Verweij, J., Loos, W., de Wit, R., de Jonge, M., Planting, A., Nooter, K., Stoter, G., and Sparreboom, A. (2001). Body-surface area-based dosing does not increase accuracy of predicting cisplatin exposure. *J Clin Oncol.*, 19(17):3733–3739.
- [8] Dikic, I. and Elazar, Z. (2018). Mechanism and medical implications of mammalian autophagy. *Nat Rev Mol Cell Biol.*, 19(6):349–364.
- [9] Dogu, Y. and Diaz, J. (2009). Mathematical model of a network of interaction between p53 and bcl-2 during genotoxic-induced apoptosis. *Biophys Chem.*, 143(1-2):44–54.
- [10] Dussmann, H., Rehm, M., Concannon, C., Anguissola, S., Wurstle, M., Kacmar, S., Voller, P., Huber, H., and Prehn, J. (2010). Single-cell quantification of bax activation and mathematical modelling suggest pore formation on minimal mitochondrial bax accumulation. *Cell Death Differ.*, 17(2):278–290.
- [11] Eisenberg, M., Kim, Y., Li, R., Ackerman, W., Kniss, D., and Friedman, A. (2011). Modeling the effects of myoferlin on tumor cell invasion. *Proc Natl Acad Sci USA*, 108(50):20078–83.
- [12] Fischer, U. and Schulze-Osthoff, K. (2005). Apoptosis-based therapies and drug targets. *Cell Death Differ.*, 12:942–961.
- [13] Gaudet, S., Spencer, S., Chen, W., and Sorger, P. (2012). Exploring the contextual sensitivity of factors that determine cell-to-cell variability in receptor-mediated apoptosis. *PLoS Comput Biol.*, 8(4):e1002482.

- [14] Gaudette, B., Dwivedi, B., Chitta, K., Poulain, S., Powell, D., Vertino, P., Leleu, X., Lonial, S., Chanan-Khan, A., Kowalski, J., and Boise, L. (2016). Low expression of pro-apoptotic Bcl-2 family proteins sets the apoptotic threshold in waldenstrom macroglobulinemia. *Oncogene.*, 35(4):479–490.
- [15] Gong, Y., Fan, Z., Luo, G., Yang, C., Huang, Q., Fan, K., Cheng, H., Jin, K., Ni, Q., Yu, X., and Liu, C. (2019). The role of necroptosis in cancer biology and therapy. *Mol Cancer.*, 18(1):100.
- [16] Huber, H., Bullinger, E., and Rehm, M. (2009). *Systems Biology Approaches to the Study of Apoptosis*. Humana Press.
- [17] Jain, H., Nor, J., and Jackson, T. (2008). Modeling the vegf-bcl-2-cxcl8 pathway in intratumoral angiogenesis. *Bull Math Biol.*, 70(1):89–117.
- [18] Jain, H., Nor, J., and Jackson, T. (2009). Quantification of endothelial cell-targeted anti-bcl-2 therapy and its suppression of tumor growth and vascularization. *Mol Cancer Ther.*, 8(10):2926–2936.
- [19] Janes, K. and Yaffe, M. (2006). Data-driven modelling of signal-transduction networks. *Nat Rev Mol Cell Biol.*, 7(11):820–828.
- [20] Jung, E., los Reyes, A., Pumares, K., and Kim, Y. (2019). Strategies in regulating glioblastoma signaling pathways and anti-invasion therapy. *PLoS One*, 14(4):e0215547.
- [21] Kim, Y., Kang, H., Powathil, G., Kim, H., Trucu, D., Lee, W., Lawler, S., and Chaplain, M. (2018a). Role of extracellular matrix and microenvironment in regulation of tumor growth and lar-mediated invasion in glioblastoma. *PLoS One*, 13:(10):e0204865.
- [22] Kim, Y., Lee, D., and Lawler, S. (2020). Collective invasion of glioma cells through OCT1 signalling and interaction with reactive astrocytes after surgery. *Phil. Trans. R. Soc. B*, 375:20190390.
- [23] Kim, Y., Lee, D., Lee, J., Lee, S., and Lawler, S. (2019a). Role of tumor-associated neutrophils in regulation of tumor growth in lung cancer development: A mathematical model. *PLoS One*, 14(1):e0211041.
- [24] Kim, Y., Lee, J., Lee, D., and Othmer, H. (2019b). Synergistic effects of bortezomib-OV therapy and anti-invasive strategies in glioblastoma: a mathematical model. *Cancers*, 11:215.
- [25] Kim, Y. and Othmer, H. (2013). A hybrid model of tumor-stromal interactions in breast cancer. *Bull Math Biol*, 75:1304–1350.
- [26] Kim, Y. and Othmer, H. (2015). Hybrid models of cell and tissue dynamics in tumor growth. *Math Bios Eng*, 12(6):1141–1156.
- [27] Kim, Y., Powathil, G., Kang, H., Trucu, D., Kim, H., Lawler, S., and Chaplain, M. (2015). Strategies of eradicating glioma cells: A multi-scale mathematical model with miR-451-AMPK-mTOR control. *PLoS One*, 10(1):e0114370.

- [28] Kim, Y. and Roh, S. (2013). A hybrid model for cell proliferation and migration in glioblastoma. *Discrete and Continuous Dynamical Systems-B*, 18(4):969–1015.
- [29] Kim, Y., Roh, S., Lawler, S., and Friedman, A. (2011a). miR451 and AMPK/MARK mutual antagonism in glioma cells migration and proliferation. *PLoS One*, 6(12):e28293.
- [30] Kim, Y., Stolarska, M., and Othmer, H. (2007). A hybrid model for tumor spheroid growth in vitro I: Theoretical development and early results. *Math. Models Methods in Appl Scis*, 17:1773–1798.
- [31] Kim, Y., Stolarska, M., and Othmer, H. (2011b). The role of the microenvironment in tumor growth and invasion. *Prog Biophys Mol Biol*, 106:353–379.
- [32] Kim, Y., Yoo, J., Lee, T., Liu, J., Yu, J., Caligiuri, M., Kaur, B., and Friedman, A. (2018b). Complex role of NK cells in regulation of oncolytic virus-bortezomib therapy. *Proc Natl Acad Sci USA*, 115(19):4927–4932.
- [33] Kueh, H., Zhu, Y., and Shi, J. (2016). A simplified Bcl-2 network model reveals quantitative determinants of cell-to-cell variation in sensitivity to anti-mitotic chemotherapeutics. *Sci Rep.*, 6:36585.
- [34] Lenhart, S. and Workman, J. (2007). *Optimal Control Applied to Biological Models*. Chapman and Hall/CRC, 1st edition edition.
- [35] Lindner, A., Concannon, C., Boukes, G., Cannon, M., Llambi, F., Ryan, D., Boland, K., Kehoe, J., McNamara, D., Murray, F., Kay, E., Hector, S., Green, D., Huber, H., and Prehn, J. (2013a). Systems analysis of bcl2 protein family interactions establishes a model to predict responses to chemotherapy. *Cancer Res.*, 73(2):519–528.
- [36] Lindner, A., Prehn, J., and Huber, H. (2013b). The indirect activation model of mitochondrial outer membrane permeabilisation (momp) initiation requires a trade-off between robustness in the absence of and sensitivity in the presence of stress. *Mol Biosyst.*, 9(9):2359–2369.
- [37] Miller, A. (2002). Body surface area in dosing anticancer agents: Scratch the surface! *J Natl Cancer Inst.*, 94(24):1822–1823.
- [38] Obied, H., Enayah, S., Ghaleb, R., and Obaid, R. (2018). The synergistic effect of cisplatin and interferon β on human lung adenocarcinoma cell line (a549). *J. Pharm.Sci and Res.*, 10(8):1939–1942.
- [39] Qiu, B., Wang, Y., Tao, J., and Wang, Y. (2012). Expression and correlation of bcl-2 with pathological grades in human glioma stem cells. *Oncol Rep.*, 28(1):155–160.
- [40] Rebecca, V., Massaro, R., Fedorenko, I., Sondak, V., Anderson, A., Kim, E., Amaravadi, R., Maria-Engler, S., Messina, J., Gibney, G., Kudchadkar, R., and Smalley, K. (2014). Inhibition of autophagy enhances the effects of the akt inhibitor mk-2206 when combined with paclitaxel and carboplatin in braf wild-type melanoma. *Pigment Cell Melanoma Res*, 27(3):465–478.

- [41] Redlarski, G., Palkowski, A., and Krawczuk, M. (2016). Body surface area formulae: an alarming ambiguity. *Sci Rep.*, 6:27966.
- [42] Rehm, M. and Prehn, J. (2013). Systems modelling methodology for the analysis of apoptosis signal transduction and cell death decisions. *Methods*, 61(2):165–173.
- [43] Reyes, A. L., Jung, E., and Kim, Y. (2015). Optimal control strategies of eradicating invisible glioblastoma cells after conventional surgery. *J. Roy Soc Interface*, 12:20141392.
- [44] Schattler, H., Kim, Y., Ledzewicz, U., los Reyes V, A., and Jung, E. (2013). On the control of cell migration and proliferation in glioblastoma. *Proceeding of the IEEE Conference on Decision and Control.*, 978-1-4673-5716-6/13:1810–1815.
- [45] Schattler, H. and Ledzewicz, U. (2015). *Optimal Control for Mathematical Models of Cancer Therapies: An Application of Geometric Methods*. Springer, 1st edition edition.
- [46] Shirin, A., Klickstein, I., Feng, S., Lin, Y., Hlavacek, W., and Sorrentino, F. (2019). Prediction of optimal drug schedules for controlling autophagy. *Sci Rep.*, 9(1):1428.
- [47] Su, Z., Yang, Z., Xie, L., DeWitt, J., and Chen, Y. (2016). Cancer therapy in the necroptosis era. *Cell Death Differ.*, 23(5):748–756.
- [48] Tavassoly, I., Parmar, J., Shajahan-Haq, A., Clarke, R., Baumann, W., and Tyson, J. (2015). Dynamic modeling of the interaction between autophagy and apoptosis in mammalian cells. *CPT Pharmacometrics Syst Pharmacol.*, 4(4):263–272.
- [49] Tindall, M. and Clerk, A. (2014). Modelling negative feedback networks for activating transcription factor 3 predicts a dominant role for mirnas in immediate early gene regulation. *PLoS Comput Biol*, 10(5):e1003597.
- [50] Tokar, T. and Ulicny, J. (2013). The mathematical model of the bcl-2 family mediated momp regulation can perform a non-trivial pattern recognition. *PLoS One.*, 8(12):e81861.
- [51] Wojton, J., Meisen, W., and Kaur, B. (2016). How to train glioma cells to die: molecular challenges in cell death. *J Neurooncol.*, 126(3):377–384.
- [52] Wurstle, M., Zink, E., Prehn, J., and Rehm, M. (2014). From computational modelling of the intrinsic apoptosis pathway to a systems-based analysis of chemotherapy resistance: achievements, perspectives and challenges in systems medicine. *Cell Death Dis.*, 5(5):e1258.